# Wild cereal grain consumption among Early Holocene foragers of the Balkans predates the arrival of agriculture

**Emanuela Cristiani[1]\*, Anita Radini[1,2], Andrea Zupancich[1], Angelo Gismondi[3], Alessia D'Agostino[3], Claudio Ottoni[1,4], Marialetizia Carra[1], Snežana Vukojičić[5], Mihai Constantinescu[6], Dragana Antonović[7], T Douglas Price[8], Dušan Borić[9,10,11]\***

[1]DANTE - Diet and Ancient Technology Laboratory, Department of Oral and Maxillofacial Sciences, Sapienza University of Rome, Rome, Italy; [2]Department of Archaeology, University of York, York, United Kingdom; [3]Laboratory of General Botany, Department of Biology, University of Rome "Tor Vergata", Rome, Italy; [4]Centre of Molecular Anthropology for Ancient DNA Studies; Department of Biology, University of Rome "Tor Vergata", Rome, Italy; [5]University of Belgrade, Faculty of Biology, Institute of Botany and Botanical Garden "Jevremovac", Belgrade, Serbia; [6]Romanian Academy, Institute for Anthropological Research "Francisc I. Rainer", Bucharest, Romania; [7]Institute of Archaeology, Belgrade, Serbia; [8]Department of Anthropology, University of Wisconsin, Madison, United States; [9]Department of Environmental Biology, Sapienza University of Rome, Rome, Italy; [10]The Italian Academy for Advanced Studies in America, Columbia University, New York, United States; [11]Department of Anthropology, New York University, New York, United States

**\*For correspondence:**
emanuela.cristiani@uniroma1.it (EC);
dusan.boric@uniroma1.it (DB)

**Competing interest:** The authors declare that no competing interests exist.

**Abstract** Forager focus on wild cereal plants has been documented in the core zone of domestication in southwestern Asia, while evidence for forager use of wild grass grains remains sporadic elsewhere. In this paper, we present starch grain and phytolith analyses of dental calculus from 60 Mesolithic and Early Neolithic individuals from five sites in the Danube Gorges of the central Balkans. This zone was inhabited by likely complex Holocene foragers for several millennia before the appearance of the first farmers ~6200 cal BC. We also analyzed forager ground stone tools (GSTs) for evidence of plant processing. Our results based on the study of dental calculus show that certain species of Poaceae (species of the genus *Aegilops*) were used since the Early Mesolithic, while GSTs exhibit traces of a developed grass grain processing technology. The adoption of domesticated plants in this region after ~6500 cal BC might have been eased by the existing familiarity with wild cereals.

## Editor's evaluation

By combining starch grains analysis from dental calculus and grinding implements, the authors demonstrate the consumption of a large variety of plants by Mesolithic foragers and Neolithic farmers in the Danube Gorges of the Balkans. The data and analyses advance debates on the intensification of plant selection prior to their strict domestication.

## Introduction

Forager knowledge and consistent use of wild cereals are still debated and poorly documented outside of the assumed centers of domestication in southwestern Asia (*Kotsakis, 2003*). For some time, it has been claimed that in the Balkans some forms of intense gathering or incipient human management of local plant and animal species might have occurred before the full-blown transition to the Neolithic (*Clarke, 1978*; *Halstead, 1996*; *Kotsakis, 2003*; *Kotzamani and Livarda, 2014*; *Srejović, 1988*; ; *y'Edynak and Fleisch, 1983*), partly due to the region's geographical proximity to the Near East. However, the hypothesis of a systematic use of wild grasses of the Poaceae family (e.g., *Aegilops* spp.; *Hordeum* spp.) during the Mesolithic remains to be verified in this region.

In southeastern Europe, specifically in its Mediterranean zone, where one would expect a greater spectrum of small seeded grasses, fruits, and nuts, forager consumption of wild cereals is well documented only at Franchthi Cave in Greece. Here, wild barley (*Hordeum* sp.) appears in the archaeobotanical record starting in the Late Upper Palaeolithic and throughout the Mesolithic, along with oat (*Avena* sp.), pulses (*Lens* sp. Mill.), bitter vetch (*Vicia ervilia* (L.) Willd.), almond (*Prunus amygdalus* Batscht), and terebinth (*Pistacia* cf. *lentiscus* L.) (*Hansen, 1991*; *Van Andel et al., 1987*). More recently, at Vlakno Cave in Croatia, starch granules of a wild species of barley (*Hordeum* spp.), along with those of oat (*Avena* spp.), were found in the dental calculus of a Mesolithic forager individual, dating to the late eight millennium cal BC burial (*Cristiani et al., 2018*).

Besides this type of evidence, data about the increase of cereal-type pollen in the Late Mesolithic (LM) come from palynological spectra from across Europe. Although the exclusive reliance on pollen evidence for inferring cultivation can be problematic, consistent evidence for interventions in the forest canopy, marked as disturbances in pollen spectra, might suggest anthropogenic activity. Due to low dispersal rates of cereal-type pollen grains as well as Cerealia-type pollens, their very presence in pollen spectra could be highly indicative of anthropic origin of disturbance phases (*Edwards, 1989*), and could be interpreted as forest clearances.

Recent methodological advances in our ability to analyse microresidues in the form of microremains along with surface modifications and microresidues on ground stone tools (henceforth GSTs) (*Barton et al., 2018*; *Dubreuil and Nadel, 2015*; *Radini et al., 2017*) have the potential to contribute to this old debate about Balkan and other prehistoric foragers' familiarity with plant species. Moreover, far from seeing foragers as passive recipients of novelties arriving from Neolithic groups at the time of agricultural transitions, there is now growing evidence of the active role of hunter–gatherers in shaping their landscape ecologies, including plant management, and manipulation of ecosystems through niche constructing (*Lombardo et al., 2020*; *Rowley-Conwy and Layton, 2011*; *Smith, 2012*).

We examine these pertinent issues in hunter–gatherer research by studying dental remains and GSTs found at Mesolithic and Neolithic sites in the Danube Gorges area of the north-central Balkans between present-day Serbia and Romania (*Figures 1 and 2*; *Figure 3*). This is one of the best researched areas of Europe regarding the Mesolithic–Neolithic transition period with more than 20 sites spanning the duration of the Epipalaeolithic through to the Mesolithic and EN (~13,000–5500 cal BC) (*Bonsall, 2008*; *Borić, 2011*; *Borić, 2016*; *Radovanović, 1996*; *Srejović, 1972*). Open-air sites began appearing in the archaeological record with the start of the Holocene warming on river terrace promontories in the vicinity of strong whirlpools, narrows, and rapids of the Danube, which facilitated intense fishing operations (*Borić, 2011*). The Early and Middle Mesolithic (~9600–7300 cal BC) deposits at many sites are damaged by later Mesolithic and Neolithic intrusions, but a number of burials have directly been dated by Accelerator Mass Spectrometry to these early phases. From the Early Mesolithic (EM) onwards, these sites became places for a continuous interment of the dead (*Borić, 2016*; *Borić et al., 2014*; *Radovanović, 1996*), thus creating a substantial mortuary record, which is in the excess of 500 individuals. Osteological collections allowed for a host of bioarchaeological analyses to be applied on this material (*Bonsall et al., 2013*; *Borić et al., 2004*; *Borić and Price, 2013*; *Mathieson et al., 2018*). Fishing seems to have remained one of the main subsistence foci throughout the Holocene, with a possible intensification during the LM (~7300–6200 cal BC), the period that saw an intense inhabitation of the area, with recognizable features in the archaeological record, such as stone-lined rectangular hearths and abundant primary burials placed as extended inhumations parallel with the Danube River. Between ~6200 and 5900 cal BC, there are clearest indications based on both material culture associations and isotope and genomic data (*Borić and Price, 2013*; *Mathieson et al., 2018*) that the local Mesolithic foragers came into contact with the first Neolithic groups appearing in this

**eLife digest** Before humans invented agriculture and the first farmers appeared in southwestern Asia, other ancient foragers (also known as hunter-gatherers) in southeastern Europe had already developed a taste for consuming wild plants. There is evidence to suggest that these foragers were intensely gathering wild cereal grains before the arrival of agriculture. However, until now, the only place outside southwestern Asia this has been shown to have occurred is in Greece, and is dated around 20,000 years ago.

In the past, researchers proposed that forager societies in the Balkans also consumed wild cereals before transitioning to agriculture. But this has been difficult to prove because plant foods are less likely to preserve than animal bones and teeth, making them harder to detect in prehistoric contexts.

To overcome this, Cristiani et al. studied teeth from 60 individuals found in archaeological sites between Serbia and Romania, which are attributed to the Mesolithic and Early Neolithic periods. Food particles extracted from crusty deposits on the teeth (called the dental calculus) were found to contain structures typically found in plants. In addition, Cristiani et al. discovered similar plant food residues on ground stone tools which also contained traces of wear associated with the processing of wild cereals.

These findings suggest that foragers in the central Balkans were already consuming certain species of wild cereal grains 11,500 years ago, before agriculture arrived in Europe. It is possible that sharing knowledge about plant resources may have helped introduce domesticated plant species in to this region as early as 6500 BC.

This work challenges the deep-rooted idea that the diet of hunter-gatherers during the Palaeolithic and Mesolithic periods primarily consisted of animal proteins. In addition, it highlights the active role the eating habits of foragers might have played in introducing certain domesticated plant species that have become primary staples of our diet today.

region, and who likely founded several new sites in this area, especially in the downstream part of the region. These documented encounters of two different cultural groups are most clearly observed at the site of Lepenski Vir (*Borić, 2016*; *Borić et al., 2018*). After ~5900 cal BC, it seems that the forager cultural specificity was lost and that various sites remained to be used as typical EN Starčevo culture villages up until ~5500 cal BC, when most of the previously used locales were abandoned.

While sources of animal protein in the diet of Mesolithic–Neolithic inhabitants of the area are well understood by now, the significance of plant foods in this region has remained less well known. Nonflaked tools such as pestles, grinders, crushers, and anvils have recently been associated with fruit, seed, and nut processing in early prehistoric and ethnographic contexts (*de Beaune, 2004*; *Dubreuil and Nadel, 2015*; *Hamon et al., 2021*; *Pardoe et al., 2019*; *Wright, 2017*). However, this category of artifacts is primarily documented from the LM onwards and only sporadically associated with earlier periods in the region of the Danube Gorges (*Antonović, 2006*; *Borić et al., 2014*; *Srejović and Letica, 1978*).

Such a lack of evidence about the role of plant foods in Mesolithic stems from very limited attempts to recover macrobotanical remains through intense sediment flotation, which has only been applied at two more recently excavated sites in this region—Schela Cladovei (*Mason et al., 1996*) and Vlasac (*Borić et al., 2014*). Despite a generally poor preservation of plant remains due to taphonomic issues, recent carpological analyses indicated a relatively wide spectrum of wild resources available to the local Mesolithic foragers. These included drupes, fruits, and berries (*Marinova et al., 2013*), among which cornelian cherry (*Cornus mas* L.), hazelnut (*Corylus avellana* L.), and elderberry (*Sambucus nigra* L.) were the most frequent taxa (*Filipović et al., 2010*; *Marinova et al., 2013*). Molecular record of *C. avellana* and *S. nigra* was also found preserved in the dental calculus of two LM individuals from Vlasac, which underwent metagenomic analysis (*Ottoni et al., 2021*). Moreover, in the study region of the Danube Gorges, at the site of Vlasac, presumed human palaeofeces contained pollen of Amaranthaceae and Cerealia (*Cârciumaru, 1978*). Evidence from the Mesolithic levels at the site of Icoana, located in the same region, has suggested local cereal cultivation (*Cârciumaru, 1973*). Pollen provides only indirect evidence for consumption and cultivation and, unfortunately, the 1960–1970s excavations of the sites in the Danube Gorges did not involve any flotation of contextual units associated

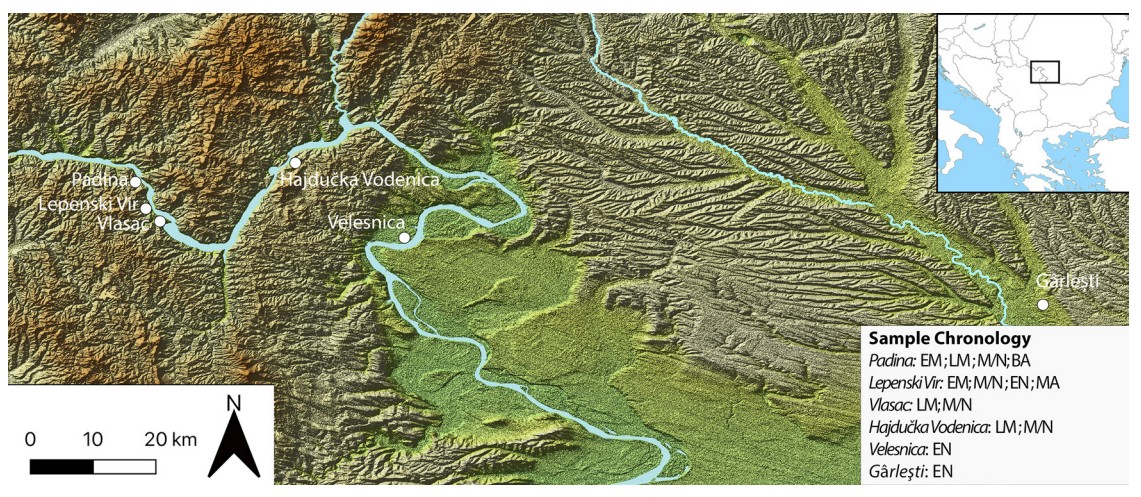

**Figure 1.** Sites in the central Balkans investigated in the article, which provided dental calculus and ground stone tools. EM = Early Mesolithic; LM = Late Mesolithic; M/N = Mesolithic-Neolithic; EN = Early Neolithic; BA = Bronze Age; MA = Medieval.

with burning in domestic contexts. While the extensive program of flotation at the site of Vlasac in the course of more recent work (2006–2009) has not led to the discovery of macrobotanical remains of wild or domesticated cereal grains, it should also be emphasized that the new excavations at this site have taken place in a marginal, upslope part of the site with little or no evidence of domestic features associated with burning that might have preserved macrobotanical remains (**Borić et al., 2014**). More recently, starch granules identified in dental calculus within a sample of 12 individuals provided evidence for the consumption of domesticated cereals at the site of Vlasac during the LM (**Cristiani et al., 2016**).

Hence, plant debris recovered in human dental calculus constitute the most reliable line of evidence to unveil the role of plants in the local forager diets. Our previous pilot study has provided the first evidence of domesticated cereal grains and plant food consumption in the LM from the analyses of dental calculus (**Cristiani et al., 2016**). Based on a more robust sample of human dental calculus, which now also involves numerous EM individuals not included in our previous study, and complementary functional evidence from the most conspicuous assemblage of Mesolithic GSTs from the Danube Gorges area, the present study details forager use of certain species of the Triticeae tribe and other plant foods in the region already since ~9500 cal BC.

## Results

### Dental calculus

Starch granules were almost ubiquitous in the analyzed individuals and many of them were found still in part associated with dental calculus remains. Six morphotypes have been retrieved in this study (**Table 1**, **Table 2**, **Table 3**). We have not attempted the identification of starch granules less than 5 μm to avoid misinterpretation of transitory and small storage starch granules (**Haslam, 2004**).

### Type I

Size, shape, morphology, and bimodal distribution that characterize granules of this type are encountered in Europe only in the members of the plant tribe Triticeae (Poaceae family) and considered diagnostic features for taxonomic identification (**Henry and Piperno, 2008**; **Stoddard, 1999**; **Yang and Perry, 2013**). Such distribution involves the presence of large granules (A-Type), mostly with a clear, round to suboval in 2D shape, ranging between 21.1 and 62.7 μm in maximum dimensions (mean size of 41.9 μm), lenticular 3D shape with equatorial groove always visible, a central hilum and high density of deep lamellae concentrated in the mesial part; and small granules (B-Type) with round/suboval shapes, a central hilum, generally smaller than 10 μm (**Geera et al., 2006**; **Stoddard, 1999**; **Yang and Perry, 2013**). A-Type granules possess diagnostic features while smaller B-Type granules

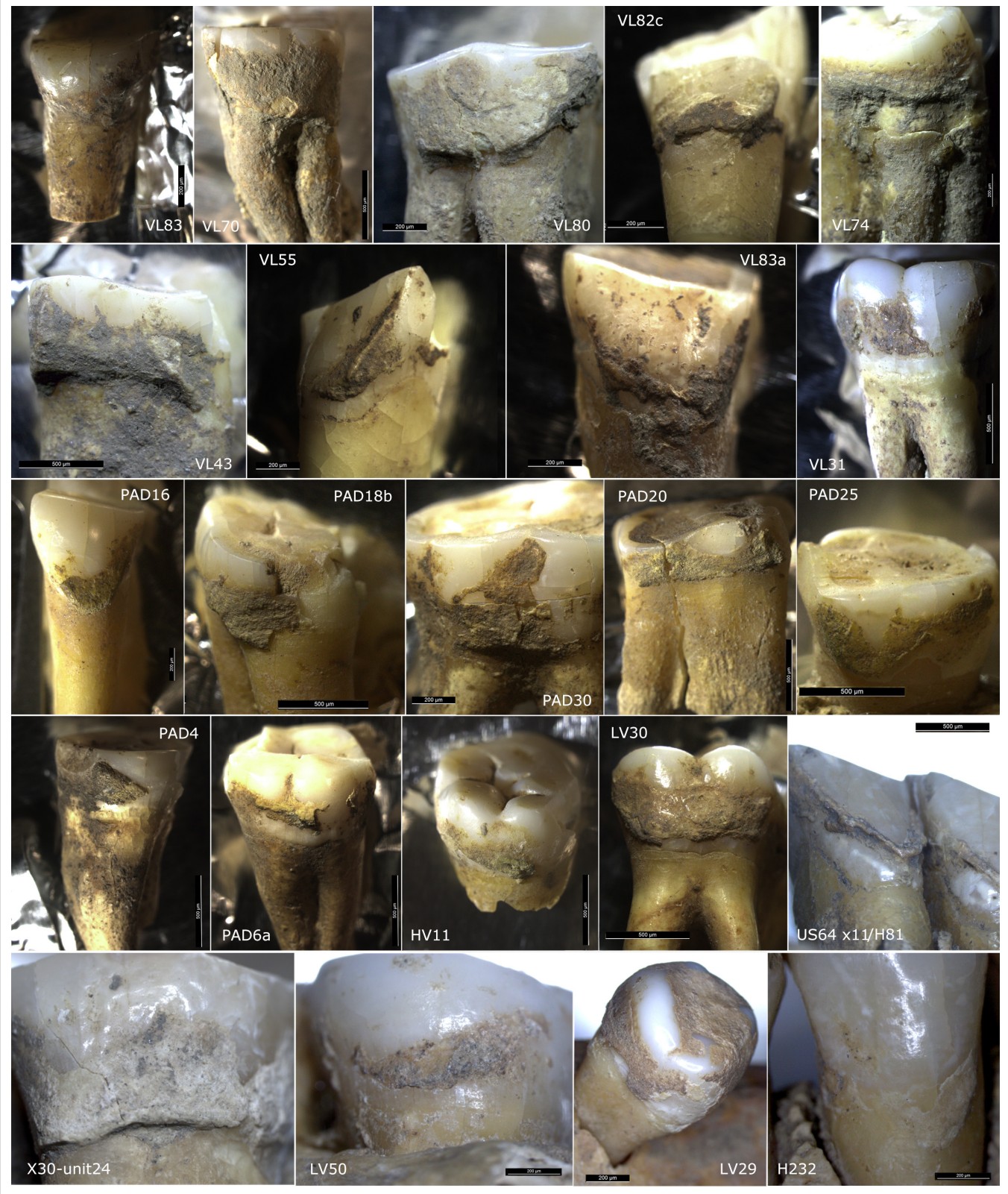

**Figure 2.** Studied teeth photographed under the microscope before dental calculus sampling.

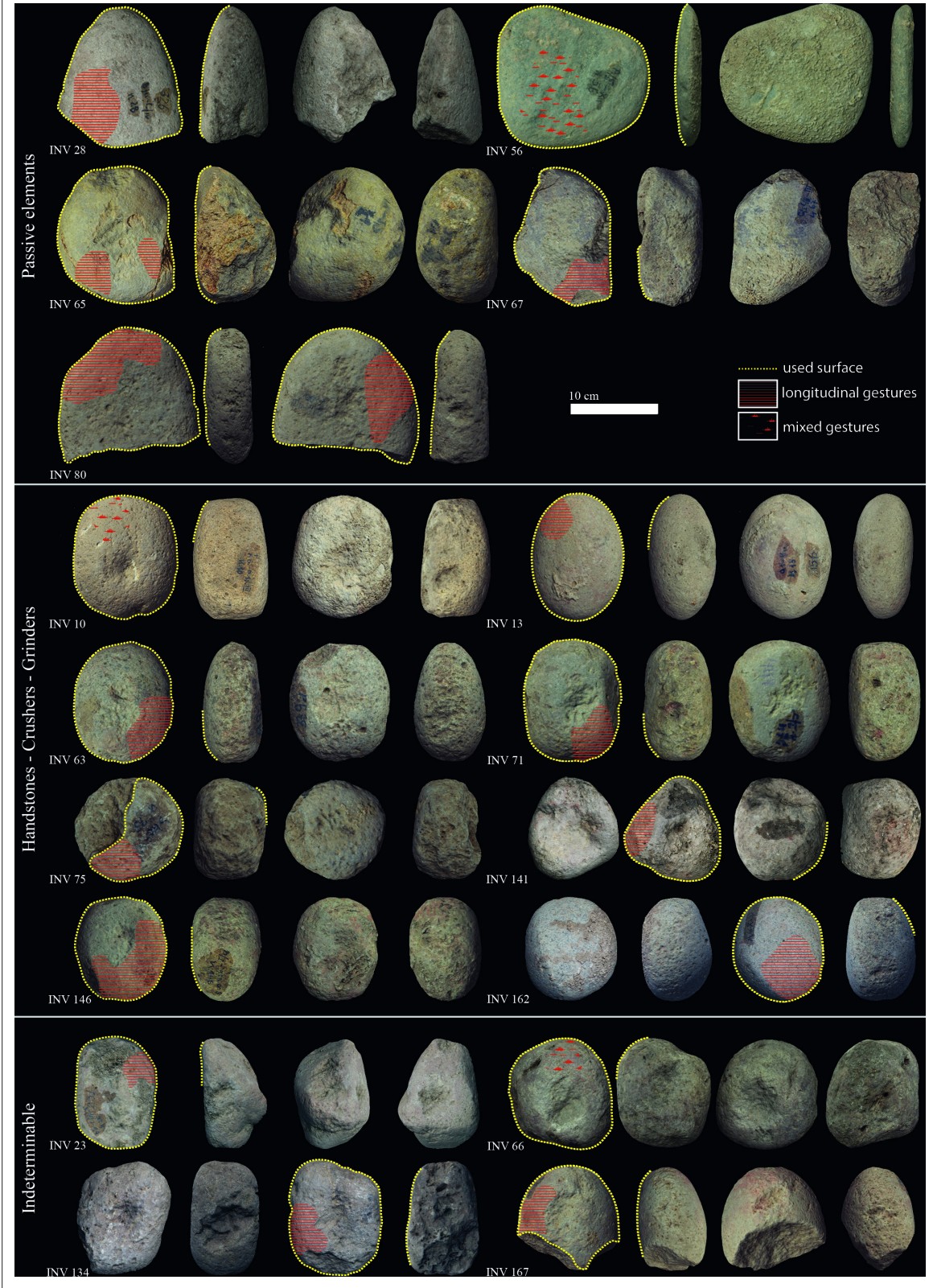

**Figure 3.** Late Mesolithic ground stone tools from the site of Vlasac featuring use-wear traces and residues related to plant food processing.

**Table 1.** Details of dental calculus sampled for the study ($n = 60$).

*No stable isotope values are currently available for this individual in order to correct the obtained radiocarbon date for the reservoir effect, and the calibrated range should probably be considered too old for its actual age, likely being 200–500 years younger. All calibrated ranges have end points rounded outwards to 5 years. The dates were individually calibrated using OxCal 4.4 and IntCal 20 (*Reimer et al., 2020*).

| Site | Burial no. | Period attribution | AMS dates | | | | Calculus location | | |
|---|---|---|---|---|---|---|---|---|---|
| | | | Lab code and source | ¹⁴C age (BP) | Reservoir effect corrected age (BP) | 95.4% confidence, cal BC | Tooth | Surface | Weight (mg) |
| Padina | PAD20 | Early Meso | | | | | 17 | Buccal | 9.6 |
| Padina | PAD25 | Early Meso | | | | | 38 | Buccal | 9.59 |
| Padina | PAD15 | Early Meso | OxA-17145 (*Borić, 2011*) | 9310 ± 44 | 8870 ± 63 | 8240–7770 | 38 | Lingual | 9.58 |
| Padina | PAD16a | Early Meso | PSU-2407 (*Mathieson et al., 2018*) | 9340 ± 35 | 8907 ± 66 | 8275–7815 | 34 | Buccal | 9.62 |
| Padina | PAD18b | Early Meso | PSU-2376 (*Mathieson et al., 2018*) | 9715 ± 40 | 9424 ± 55 | 9115–8550 | 48 | Lingual | 9.67 |
| Padina | PAD9 | Early Meso | AA-57771 (*Borić, 2011*) | 9920 ± 100 | 9480 ± 110 | 9225–8495 | 42, 46 | Lingual | 9.59 |
| Padina | PAD11 | Early Meso | OxA-16938 (*Borić, 2011*) | 9665 ± 54 | 9225 ± 70 | 8620–8290 | 27 | Lingual | 9.57 |
| Padina | PAD12 | Early Meso | BM-1146 (*Borić, 2011*) | 9331 ± 58 | – | 8750–8350 | 27 | Lingual | 9.76 |
| Padina | PAD17 | Early Meso | PSU-2375 (*Mathieson et al., 2018*) | 9505 ± 35 | 9105 ± 62 | 8540–8230 | 25 | Buccal | 9.64 |
| Lepenski Vir | LV50 | Early Meso | BA-10651 (*Borić et al., 2018*) | 9455 ± 38 | 9082 ± 62 | 8540–8020 | 35 | Buccal | 9.99 |
| Lepenski Vir | LV20 | Early Meso | OxA-39629 (this paper) | 10,268 ± 38 | 9928 ± 58 | 9740–9270 | 48 | Lingual | 9.73 |
| Padina | PAD26 | Early Meso | | | | | 14 | Buccal | 9.58 |
| Padina | PAD6 | Early Meso | | | | | 47 | Lingual | 9.65 |

*Table 1 continued on next page*

*Table 1 continued*

| Site | Burial no. | Period attribution | | AMS dates | | | Calculus location | | |
|---|---|---|---|---|---|---|---|---|---|
| Padina | PAD2 | Late Meso | BM-1143 (*Borić, 2011*) | 7738 ± 51 | – | 6650–6465 | 36 | Lingual | 9.68 |
| Hajdučka Vodenica | HV25/26 | Late Meso | | | | | 44 | Buccal | 9.60 |
| Hajdučka Vodenica | HV29 | Late Meso | AA-57774 (*Borić, 2011*) | 8151 ± 60 | 7711 ± 75 | 6690–6425 | 48 | Lingual | 10.72 |
| Hajdučka Vodenica | HV8 | Late Meso | OxA-13613 (*Borić, 2011*) | 8456 ± 37 | 8016 ± 58 | 7075–6695 | 48 | Buccal | 9.61 |
| Hajdučka Vodenica | HV11 | Late Meso | | | | | 48 | Buccal | 9.71 |
| Hajdučka Vodenica | HV profil A | Late Meso | | | | | 27 | Buccal | 9.70 |
| Hajdučka Vodenica | HV30 | Late Meso | | | | | 27 | Buccal | 9.56 |
| Vlasac | VL82c | Late Meso | BRAMS-2588 (*Jovanović et al., 2021a*) | 8035 ± 28 | 7595 ± 53 | 6590–6270 | 42 | Buccal | 9.68 |
| Vlasac | VL2 | Late Meso | | | | | 14 | Buccal | 9.54 |
| Vlasac | VL80a | Late Meso | | | | | 26 | Lingual | 9.84 |
| Vlasac | VL55 | Late Meso | BRAMS-2583 (*Jovanović et al., 2021b*) | 8377 ± 29 | 7837 ± 63 | 7035–6500 | 33 | Lingual | 9.64 |
| Vlasac | VL74 | Late Meso | BRAMS-2587 (*Jovanović et al., 2021b*) | 8149 ± 28 | * | 7315–7055* | 28 | Lingual | 9.70 |
| Vlasac | VL83 | Late Meso | OxA-5826 (*Borić, 2011*) | 8200 ± 90 | 7760 ± 100 | 7030–6420 | 24 | Lingual | 9.62 |
| Vlasac | VL43 | Late Meso | | | | | 27 | Lingual | |
| Vlasac | VL31 | Late Meso | AA-57777 (*Borić, 2011*) | 8196 ± 69 | 7756 ± 82 | 6900–6430 | 26 | Buccal | 9.58 |
| Vlasac | VL45 | Late Meso | AA-57778 (*Borić et al., 2004*) | 8117 ± 62 | 7677 ± 77 | 6655–6400 | 38 | Buccal | 9.50 |
| Vlasac | VL70 | Late Meso | | | | | 17 | Buccal | 10.42 |

*Table 1 continued*

| Site | Burial no. | Period attribution | | AMS dates | | | Calculus location | | |
|---|---|---|---|---|---|---|---|---|---|
| Vlasac | VL79 | Late Meso | BRAMS-2448 (*Jovanović et al., 2021b*) | 8005 ± 29 | 7565 ± 54 | 6565–6250 | 16 | Buccal | 9.60 |
| Vlasac | U44 | Late Meso | | | | | 27 | Buccal | 9.96 |
| Vlasac | H232 | Late Meso | OxA-20702 (*Borić, 2011*) | 7725 ± 40 | | 6640–6470 | 28 | Lingual | 9.92 |
| Vlasac | H317 | Late Meso | PSU-2381 (*Mathieson et al., 2018*) | 8110 ± 35 | 7625 ± 71 | 6645–6270 | 26, 36 | Lingual | 9.73 |
| Vlasac | U115 | Late Meso | | | | | 28 | Buccal | 9.95 |
| Vlasac | U326 | Late Meso | PSU-2382 (*Mathieson et al., 2018*) | 8045 ± 30 | 7728 ± 51 | 6645–6465 | 17 | Buccal | 9.94 |
| Vlasac | U326 | Late Meso | PSU-2382 (*Mathieson et al., 2018*) | 8045 ± 30 | 7728 ± 51 | 6650–6460 | 1, 2 | Buccal | 9.1 |
| Vlasac | U64 x.11/H81 | Late Meso | OxA-20762 (*Borić, 2011*) | 8125 ± 45 | 7685 ± 64 | 6645–6430 | 20, 26, 27, 29, 30, 31 | Lingual | 9.92 |
| Vlasac | H341 | Late Meso | | | | | 1 | Buccal | 10.12 |
| Vlasac | VL48 | Late Meso | | | | | 34 | Lingual | 10.06 |
| Vlasac | U222 x.18 | Late Meso | | | | | 2 | Buccal | 9.54 |
| Lepenski Vir | LV28 | Meso-Neo | | | | – | 43 | Buccal | 9.58 |
| Lepenski Vir | LV79a | Meso-Neo | OxA-25091 (*Bonsall, 2008*) | 7605 ± 38 | 7119 ± 74 | 6220–5805 | 33 | Buccal | 9.69 |
| Hajdučka Vodenica | HV16 | Meso-Neo | | | | | 36 | Lingual | 9.54 |
| Hajdučka Vodenica | HV19 | Meso-Neo | | | | | 37 | Buccal | 9.58 |
| Hajdučka Vodenica | HV13 | Meso-Neo | AA-57773 (*Borić, 2011*) | 7435 ± 70 | 6995 ± 83 | 6020–5720 | 17 | Lingual | 9.56 |

*Table 1 continued on next page*

*Table 1 continued*

| Site | Burial no. | Period attribution | Lab code (reference) | AMS dates | | cal BC range | Calculus location | | |
|---|---|---|---|---|---|---|---|---|---|
| Padina | PAD4 | Meso-Neo | AA-57769 (**Ottoni et al., 2021**) | 7518 ± 72 | 7078 ± 85 | 6080–5745 | 48 | Buccal | 9.73 |
| Padina | PAD5 | Meso-Neo | AA-57770 (**Borić, 2011**) | 7598 ± 72 | 7158 ± 85 | 6230–5845 | 15 | Buccal | 8.10 |
| Vlasac | U24 x.30 | Meso-Neo | | | | | 32 | Lingual | 9.97 |
| Vlasac | H53 | Meso-Neo | OxA-16544 (**Borić et al., 2014**) | 7035 ± 40 | – | 6015–5805 | 3, 28, 29 | Lingual | 10.04 |
| Lepenski Vir | LV4 | Early Neo | | | | | 33 | Buccal | 9.64 |
| Lepenski Vir | LV73 | Early Neo | BA-10652 (**Borić et al., 2018**) | 7265 ± 30 | 6973 ± 48 | 5980–5735 | 34 | Buccal | 9.69 |
| Lepenski Vir | LV8 | Early Neo | AA-58319 OxA-25207 (**Borić et al., 2018**) | 6825 ± 51 7097 ± 36 | 6690 ± 54 6984 ± 39 | 5715–5520 5985–5750 | 44 | Lingual | 9.69 |
| Lepenski Vir | LV32A | Early Neo | OxA-5828 (**Bonsall et al., 2013**) | 7270 ± 90 | 7032 ± 95 | 6065–5730 | 42, 43, 36 | Buccal | 9.77 |
| Lepenski Vir | LV17 | Early Neo | AA-58320 (**Borić et al., 2018**) | 7007 ± 48 | 6787 ± 53 | 5775–5565 | 15 | Lingual | 9.10 |
| Padina | PAD30 | Bronze Age | PSU-2379 | | | 2140–1765 | 47 | Buccal | 9.69 |
| Velesnica | 2A | Early Neo | OxA-19191 (**Bonsall, 2008**) | 7409 ± 38 | 7196 ± 47 | 6220–5930 | 8 | Lingual | 9.74 |
| Velesnica | 2D | Early Neo | OxA-19210 (**Bonsall, 2008**) | 7327 ± 38 | 7183 ± 42 | 6215–5925 | 9 | Lingual | 7.2 |
| Gârleşti | | Early Neo | | | | | 2 | Lingual | 8.22 |
| Lepenski Vir | LV30 | Medieval | OxA-25218 (**Bonsall, 2008**) | 427 ± 23 | | AD1440–1490 | 16 | Lingual | 9.67 |

**Table 2.** Details of the microdebris (starch granules and other microremains) found in the archaeological dental calculus samples (PO = pollen; W = wood; Ch = microcharcoal/burnt debris; Gr = grit; P = phytoliths; FE = feathers; FI = fibers; FU = fungi; S = smoke) (n = 51).

| | Site | Burial label | Chronocultural attribution | Type I Triticeae | Type II Aveneae | Type III Paniceae | Type IV Fabeae | Type V Fagaceae | Type VI Cornaceae | Indet. | Other |
|---|---|---|---|---|---|---|---|---|---|---|---|
| 1 | Padina | PAD20 | Early Meso | | | 7 | 4 | | | 3 | 1P/10FI/2FE/2W/1Ch/Gr |
| 2 | Padina | PAD25 | Early Meso | | | 1 | | | | | 1P/1Ch/Gr |
| 3 | Padina | PAD15 | Early Meso | >100 | 4 | | 6 | | | | 1P/4PO/1W/1Ch/Gr |
| 4 | Padina | PAD16a | Early Meso | | | 20 | | | | 3 | 1P/10FI/8Ch/Gr |
| 5 | Padina | PAD9 | Early Meso | >200 | | | | | | | 2FI/2FE |
| 6 | Padina | PAD11 | Early Meso | >100 | | 8 | | 1 | | | |
| 7 | Padina | PAD12 | Early Meso | 36 | | 12 | | 1 | | | |
| 8 | Lepenski Vir | LV50 | Early Meso | | | | 1 | | | 1 | |
| 9 | Lepenski Vir | LV20 | Early Meso | 4 | | | | | | 1 | 1FI;S |
| 10 | Padina | PAD2 | Late Meso | 13 | 1 | 7 | 1 | | | | 1PO/1FI/2FE/2W/1Ch/Gr |
| 11 | Hajdučka Vodenica | HV25/26 | Late Meso | 5 | | 15 | | | | 1 | 2P/1PO/2FI/3FE/1Ch/3FU/Gr |
| 12 | Hajdučka Vodenica | HV29 | Late Meso | | 5 | | 3 | | | 5 | 3P/1PO/1FI/1FE/13W/1FU/Gr |
| 13 | Hajdučka Vodenica | HV8 | Late Meso | | 1 | 1 | | | | | |
| 14 | Hajdučka Vodenica | HV11 | Late Meso | >100 | 8 | 1 | | | | | |
| 15 | Hajdučka Vodenica | HV profil A | Late Meso | | | 14 | | | | 1 | 3PO/1FI/2FE/3W/1Ch/1FU/Gr |
| 16 | Hajdučka Vodenica | HV30 | Late Meso | | | 1 | | | | | |
| 17 | Vlasac | U222 x.18 | Late Meso | | | | 2 | | | | 1P |
| 18 | Vlasac | U326 | Late Meso | >60 | | | | | | 1 | 1P |
| 19 | Vlasac | VL82c | Late Meso | 4 | 23 | 7 | 2 | | | 5 | 1P/3PO/1FE/1W/1Ch/Gr |
| 20 | Vlasac | VL2 | Late Meso | 2 | 12 | | | | | | 2P/1PO/2FI/1FE/2FU/Gr |
| 21 | Vlasac | VL80a | Late Meso | 3 | 15 | 5 | 5 | 1 | | | 1P/4FE/1W/1Ch/1FU/Gr |
| 22 | Vlasac | VL55 | Late Meso | | | 6 | | | | 1 | 1P/1PO/1FI/1W/1FU/Gr |

*Table 2 continued on next page*

*Table 2 continued*

| | Site | Burial label | Chronocultural attribution | Type I Triticeae | Type II Aveneae | Type III Paniceae | Type IV Fabeae | Type V Fagaceae | Type VI Cornaceae | Indet. | Other |
|---|---|---|---|---|---|---|---|---|---|---|---|
| 23 | Vlasac | VL74 | Late Meso | | | 1 | | | | 1 | 1P/3PO/17FI/1Ch/Gr |
| 24 | Vlasac | VL83 | Late Meso | 6 | | 8 | 1 | | | | 1P/1PO/1FE/2W/5Ch/Gr |
| 25 | Vlasac | VL43 | Late Meso | >200 | 12 | 1 | | 1 | | 2 | 2FE/1W/2Ch/1FU/Gr |
| 26 | Vlasac | VL31 | Late Meso | | | 18 | 3 | | | 3 | 4PO/5FE/1Ch/2FU/Gr |
| 27 | Vlasac | VL45 | Late Meso | 23 | 8 | 20 | | 1 | | 14 | 1PO/2FE/2Ch |
| 28 | Vlasac | VL70 | Late Meso | 3 | | 3 | | 1 | 14 | | 4P/1FI/7Ch |
| 29 | Vlasac | VL79 | Late Meso | | | | | | | | 1P/2FI |
| 30 | Vlasac | U44 | Late Meso | 3 | | | 2 | | | | |
| 31 | Vlasac | H232 | Late Meso | <100 | 4 | 1 | | 1 | | 4 | 1PO/1FE |
| 32 | Vlasac | U115 | Late Meso | | | | | | | | |
| 33 | Vlasac | U64 x.11 | Late Meso | >200 | 10 | 32 | | | | 4 | 2P/4FI/2FE/3Ch/1FU |
| 34 | Vlasac | H341 | Late Meso | 1 | | | | | | | |
| 35 | Lepenski Vir | LV28 | Meso-Neo | 4 | | 3 | 1 | | | 6 | 2P/2PO/1FE/2W/4Ch/Gr |
| 36 | Hajdučka Vodenica | HV16 | Meso-Neo | >200 | | | | | | | 1FU |
| 37 | Hajdučka Vodenica | HV19 | Meso-Neo | 1 | | | | | | | 1FE |
| 38 | Hajdučka Vodenica | HV13 | Meso-Neo | | 1 | | | | | | |
| 39 | Padina | PAD4 | Meso-Neo | | | | | 1 | | 7 | 2PO/1FI/3FE/4W/2Ch/1FU/Gr |
| 40 | Padina | PAD5 | Meso-Neo | 6 | | | | | | | |
| 41 | Vlasac | U24 x.30 | Meso-Neo | | | | | | 10 | | 1P/2Ch |
| 42 | Vlasac | H53 | Meso-Neo | 22 | >200 | 5 | | | | | 1FE/1W |
| 43 | Lepenski Vir | LV4 | Early Neo | 1 | 3 | | | | | 1 | |
| 44 | Lepenski Vir | LV73 | Early Neo | 12 | 9 | | | | | 1 | 7P/2PO/17FI/1FE/3FU/Gr |
| 45 | Lepenski Vir | LV8 | Early Neo | 11 | | | 4 | | | | 1W |

*Table 2 continued on next page*

*Table 2 continued*

| Site | Burial label | Chronocultural attribution | Type I Triticeae | Type II Aveneae | Type III Paniceae | Type IV Fabeae | Type V Fagaceae | Type VI Cornaceae | Indet. | Other |
|---|---|---|---|---|---|---|---|---|---|---|
| 46 Lepenski Vir | LV32A | Early Neo | | 8 | >200 | | | | | 1P/2FE |
| 47 Lepenski Vir | LV17 | Early Neo | 14 | | | 5 | | | | |
| 48 Velesnica | 2A | Early Neo | | | | | | | | 4FU |
| 49 Velesnica | 2D | Early Neo | 12 | | | | | | | 2P/2FU |
| 50 Gârlești | | Early Neo | 1 | | | | | | | 1P/1Ch |
| 51 Lepenski Vir | LV30 | Medieval | | | 12 | 4 | | | | 1P/1PO/4Ch/1FU/Gr |
| **Total** | | | >1446 | 324 | >409 | 43 | 8 | | 24 | 284 |

**Table 3.** Late Mesolithic ground stone tools from the site of Vlasac.

| Inv. no. | Archaeological context | Shape | Tool type | Length (cm) | Width (cm) | Thickness (cm) | Weight (g) | Volume (cm³) | State of preservation | PDM | Micropolish description | Micropolish location | Microstriation description | Microstriation orientation | Cristal grain modification | Gesture |
|---|---|---|---|---|---|---|---|---|---|---|---|---|---|---|---|---|
| 10 | a1-III | Subangular | Handstone/grinder | 12.7 | 10.5 | 7.88 | 1542 | 645 | Preserved | Light soil concretion | Smooth and domed | High microtopographies | Short narrow with a matt bottom | Unidirectional | Y | Mixed |
| 13 | a1-VIII | Round | Handstone//grinder | 11 | 8.16 | 5.68 | 680 | 287 | Preserved | None | Smooth and domed with sporadic pits | High and low microtopographies | NA | NA | N | Longitudinal |
| 23 | BV/C/IV-X | Subangular | Indeterminable | 11.7 | 8.79 | 8.76 | 823 | 367 | Preserved | Light soil concretion | Rough to smooth with domed and flat spots | High and low microtopographies | NA | NA | N | Longitudinal |
| 28 | BIII-C/V | Oval | Passive base | 13.3 | 11.8 | 7.18 | 1283 | 499 | Fractured | None | Smooth | High microtopographies | NA | NA | Y | Longitudinal |
| 56 | A/II-XIII | Round | Passive base | 15.9 | 14.8 | 7.7 | 1038 | 368 | Preserved | Heavy surface concretion on one surface | Smooth domed and flat | High microtopographies | NA | NA | Y | Mixed |
| 63 | b/17-XV | Round | Handstone/grinder | 8.4 | 6.6 | 4.47 | 403 | 141 | Preserved | Light surface abrasion | Rough to smooth with reticulated and flat spots | High and low microtopographies | Narrow with a matt bottom | NA | N | Longitudinal |
| 65 | C/I-VI | Round | Passsive base | 100 | 84.3 | 55.5 | 680 | 298 | Broken | Fractures | Smooth domed and reticulated | High microtopographies | NA | Mixed | N | Longitudinal |
| 66 | C/I II/V | Round | Indeterminable | 9.5 | 8.5 | 7.6 | 1170 | 437 | Preserved | None | Smooth domed and cratered | High and low topographies | NA | NA | Y | Mixed |
| 67 | C/I-C/II-III | Subangular | Passive base | 10.8 | 8.4 | 5.91 | 633 | 253 | Broken | Light soil concretion and surface abrasion | Smooth and reticulated | High microtopographies | Short and deep with a matt bottom | Unidirectional | N | Longitudinal |
| 71 | C/I-V | Round | Handstone/grinder | 72.1 | 57.9 | 45.9 | 309 | 119 | Preserved | None | Smooth domed to flat | High microtopographies | Long and shallow with a polished bottom | Mixed | Y | Longitudinal |
| 75 | b/18V | Subangular | Handstone/grander | 5.53 | 5.29 | 3.45 | 137 | 57 | Broken | Fractures | Smooth and domed | High and low microtopographies | Short narrow with a polished bottom | Unidirectional | N | Longitudinal |
| 80 | C/II-II/6 | Ovate | Passive base | 9.85 | 8.35 | 3.72 | 547 | 225 | Broken | None | Smooth domed | High microtopographies | Short and narrow with a matt bottom | Unidirectional | Y | Longitudinal |
| 134 | b/V3-XII | Subangular | Indeterminable | 12.2 | 9.57 | 8.13 | 1433 | 565 | Preserved | None | Smooth domed | High and low microtopographies | Short deep with a matt bottom | Unidirectional | Y | Longitudinal |
| 141 | B/I 0-8.9 | Round | Handstone/grinder | 10.8 | 9.92 | 9.22 | 370 | NA | Preserved | Soil concretion | Smooth domed and flat | High microtopographies | NA | NA | Y | Longitudinal |

*Table 3 continued on next page*

Table 3 continued

| Inv. no. | Archaeological context | Shape | Tool type | Length (cm) | Width (cm) | Thickness (cm) | Weight (g) | Volume (cm³) | State of preservation | PDM | Micropolish description | Micropolish location | Microstriation description | Microstriation orientation | Cristal grain modification | Gesture |
|---|---|---|---|---|---|---|---|---|---|---|---|---|---|---|---|---|
| 146 | B/I-below hearth 9 | Round | Handstone/grinder | 6.65 | 5.7 | 4.66 | 275 | 106 | Preserved | Light soil concretion | Smooth domed and reticulated | High microtopographies | Short narrow with a matt bottom | Unidirectional | N | Longitudinal |
| 162 | A/16X | Round | Handstione/grinder | 10.6 | 9.3 | 7.52 | 1143 | 424 | Preserved | None | Smooth domed | High and low microtopographies | Shirt narrow with a matt bottom | Unidirectional | Y | Longitudinal |
| 167 | a/15-VII | Round | Indeterminable | 8.94 | 8.82 | 5.65 | 611 | 241 | Broken | Light surface abrasion | Rough granular and domed | High and low topographies | NA | NA | Y | Orthogonal |

**Table 4.** Summary statistics of the length (μm) of wild grass grains and domestic cereal starch granules.

IQR, interquartile range.

| Species | Min. | Max. | Mean | Median | St. Dev. | Range | IQR |
|---|---|---|---|---|---|---|---|
| A. caudata | 5.29 | 59.3 | 21.6 | 16.7 | 15.17 | 5.29–59.33 | 26.55 |
| A. comosa | 7.95 | 34.5 | 21.5 | 21.7 | 9.78 | 7.95–34.54 | 20.09 |
| A. crassa | 13.38 | 53.7 | 35.3 | 33.7 | 11.09 | 13.38–53.69 | 19.08 |
| A. cylindrica | 8.52 | 54.0 | 24.2 | 23.7 | 13.07 | 8.52–54.05 | 21.6 |
| A. geniculata | 11.61 | 47.0 | 26.3 | 26.0 | 8.39 | 11.61–47.03 | 12.87 |
| A. neglecta recta | 10.54 | 62.7 | 35.0 | 36.2 | 14.46 | 10.54–62.71 | 26.5 |
| A. peregrina | 9.84 | 53.6 | 27.8 | 25.9 | 9.89 | 9.84–53.62 | 11.34 |
| A. speltoides tauschii | 13.25 | 40.0 | 23.5 | 22.2 | 5.93 | 13.25–39.97 | 8.39 |
| A. triuncialis | 5.60 | 50.1 | 28.2 | 28.2 | 11.24 | 5.60–50.06 | 15.18 |
| A. uniaristata | 14.35 | 62.4 | 38.2 | 39.3 | 12.87 | 14.35–62.38 | 22.83 |
| A. ventricosa | 14.10 | 40.0 | 26.3 | 25.7 | 7.44 | 14.10–40.04 | 12.77 |
| H. vulgare distichon | 5.19 | 29.6 | 19.7 | 22.2 | 8.12 | 5.19–29.59 | 8.32 |
| T. dicoccum | 6.17 | 41.5 | 16.5 | 12.8 | 8.66 | 6.17–41.55 | 14.07 |
| T. monococcum | 6.68 | 36.6 | 20.1 | 19.1 | 7.11 | 6.68–36.61 | 10.44 |

The online version of this article includes the following source data for table 4:

**Source data 1.** Summary statistics of the length of wild grass grains and domestic cereal starch granules.

are rarely diagnostic to taxa (*Yang and Perry, 2013*). However, in our archaeological population, variability in the proportion and dimension of small B-Type granules has been noticed, resulting in a unimodal granule size distribution without a clear distinction between A and B granules in some cases. Several studies (*Howard et al., 2011*; *Stoddard and Sarker, 2000*) suggested that this characteristic is common in the species of the genus *Aegilops* of the Triticeae tribe and can be attributed to both environment (*Blumenthal et al., 1995*; *Blumenthal et al., 1994*) and genetics (*Stoddard and Sarker, 2000*). A unimodal starch granule size distribution characterized by normal A-Type granules and a lack/reduced quantity of B-Type granules was also evident in our modern reference collection of local *Aegilops* species (Figure 6). Based on these observations, Type I category was further divided into two subtypes (Ia and Ib). In subtype Ia, B-Type granules are small, dimensionally uniform (up to 12 μm) and round in shape (Figure 6). Conversely, subtype Ib is characterized by a high variation in starch granule size not allowing for a distinction between A- and B-Type granule, resulting in a unimodal distribution (Figure 9).

Type I (Ia and Ib) is very common in the analyzed samples (*Table 2*), as already emphasized in our earlier study albeit in different quantities (*Cristiani et al., 2016*). These starch granules were documented, often lodged in the amyloplast, in most of the analyzed Mesolithic individuals (5 for EM, 16 for LM, 5 for M/N), and in 5 EN individuals of our population (*Table 2*). A-Type granules recovered in EM and most of the LM individuals were very large, mostly with a clear, round shape, central hilum, and high density of deep lamellae mainly concentrated in the mesial part of the granules. Based on literature (*Henry et al., 2011*; *Yang and Perry, 2013*) and our extensive experimental and statistical results on modern botanical collection (*Table 4*; *Table 5*; Figures 6, 7 and 9), we confirm that these characteristics are consistent with A-Type granules of most species of the Triticeae tribe.

## Subtype Ia

A few LM and M/N individuals (e.g., HV11 and 16, H53, 64.x11, H327, H232) yielded a combination of oval A-Type granules and uniformly small, round B-Type granules (*Figure 4m*). Our previous claims that this pattern is a recognizable feature of the domestic species of the tribe Triticeae (e.g., *Triticum* spp./*Hordeum* spp.) (Figure 6 and 9; *Cristiani et al., 2016*) are now further supported by a

**Table 5.** Pairwise Wilcoxon test performed on the length distribution of modern starches from *Aegilops*, *Hordeum*, and *Triticum* species (p value: not significant/ns >0.05; *<0.05; **<0.01; ***<0.001).

Significance analysis performed on the length distribution of modern starches from *Aegilops*, *Hordeum*, and *Triticum* species (p value: not significant/ns >0.05; *<0.05; **<0.01; ***<0.001).

| | A. caudata | A. comosa | A. crassa | A. cylindrica | A. geniculata | A. neglecta recta | A. peregrina | A. speltoides tauschii | A. triuncialis | A. uniaristata | A. ventricosa | H. vulgare distichon | T. dicoccum |
|---|---|---|---|---|---|---|---|---|---|---|---|---|---|
| A. comosa | ns | — | — | — | — | — | — | — | — | — | — | — | — |
| A. crassa | *** | *** | — | — | — | — | — | — | — | — | — | — | — |
| A. cylindrica | ns | ns | *** | — | — | — | — | — | — | — | — | — | — |
| A. geniculata | ns | * | *** | ns | — | — | — | — | — | — | — | — | — |
| A. neglecta recta | *** | *** | ns | *** | *** | — | — | — | — | — | — | — | — |
| A. peregrina | * | ** | ** | ns | ns | ** | — | — | — | — | — | — | — |
| A. speltoides tauschii | ns | ns | *** | ns | ns | *** | * | — | — | — | — | — | — |
| A. triuncialis | * | ** | ** | ns | ns | * | ns | * | — | — | — | — | — |
| A. uniaristata | *** | *** | ns | *** | *** | ns | *** | *** | *** | — | — | — | — |
| A. ventricosa | ns | * | *** | ns | ns | *** | ns | ns | ns | *** | — | — | — |
| H. vulgare distichon | ns | ns | *** | ns | *** | *** | *** | * | *** | *** | *** | — | — |
| T. dicoccum | ns | * | *** | ** | *** | *** | *** | *** | *** | *** | *** | ns | — |
| T. monococcum | ns | ns | *** | ns | *** | *** | *** | * | *** | *** | *** | ns | * |

The online version of this article includes the following source data for table 5:

**Source data 1.** Length of modern starch granules of *Aegilops*, *Hordeum*, and *Triticum* species.

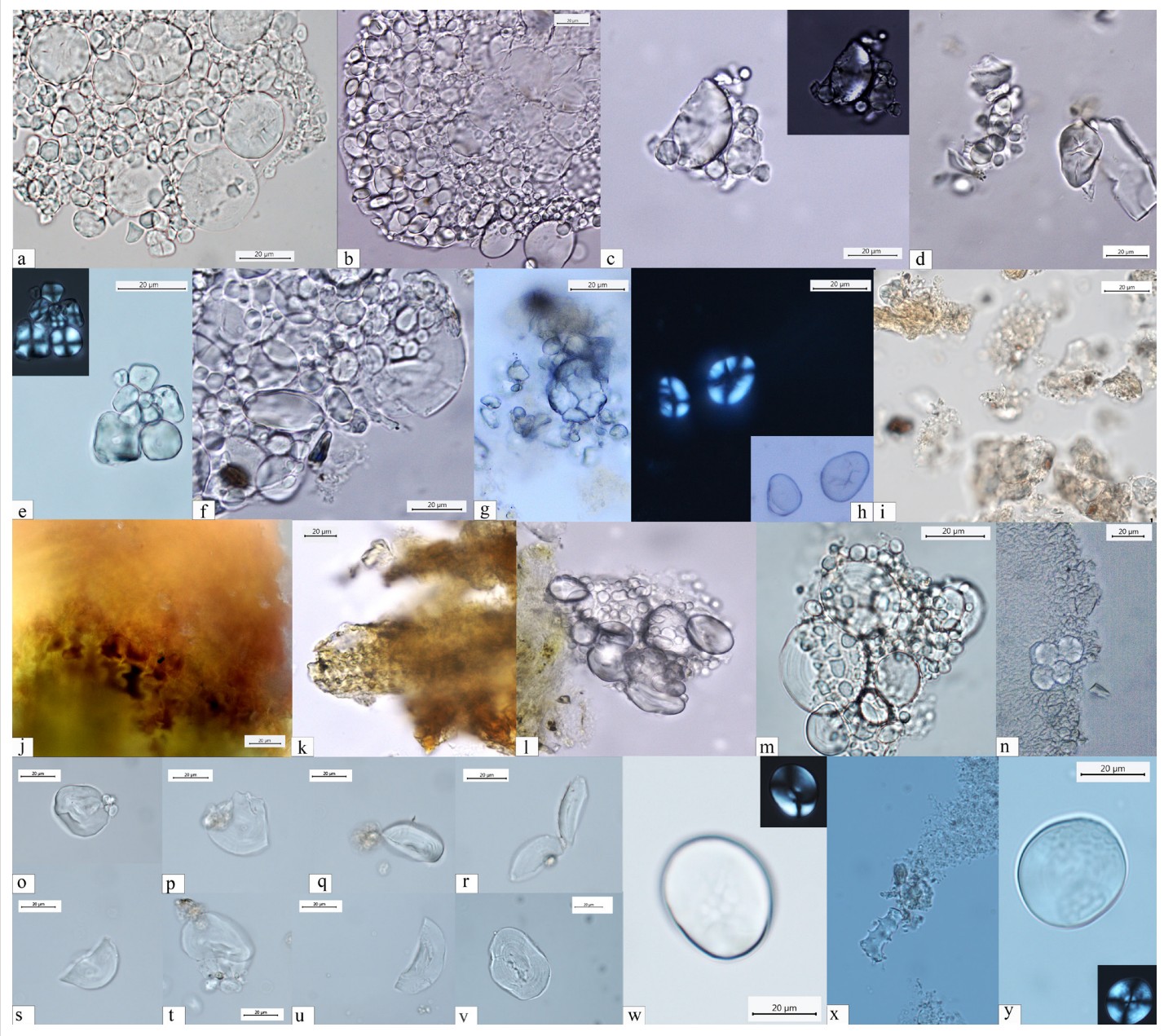

**Figure 4.** Starch granules from Mesolithic and Neolithic dental calculus. *Early Mesolithic*: (**a**) Type Ib (PAD11); (**b**) Type Ib (PAD9); (**c**) Type Ib (PAD11); (**d**) Type V (PAD12); (**e**) Type III (PAD11); (**f**) Type Ib (PAD15); *Late Mesolithic*: (**g**) Type II (VL82c); (**h**) Type IV (VL31); (**i**) Type VI (VL70); (**j, k**) multicellular structures of long cells embedded in dental calculus (HV25/26, VL70); (**l**) Type Ia (HV11); *Mesolithic–Neolithic*: (**m**) Type Ia (HV16). *Neolithic*: (**n**) Type III (LV32a); (**o–v**) damaged Type I granules (A-Type granules) (VEL-2D); (**w**) Type I (A-Type granule) (VEL-2D); (**x**) single dendritic cell (Gârleşti); (**y**) Type I (A-Type granule) (VEL-2A).

new morphometric analysis of both domestic and wild Triticeae species (Figure 9). Moreover, in the same individuals, A-Type granules could show cratered appearance (*Figure 4*). Similarly, in the EN individuals, lenticular and oval/suboval A-Type granules with equatorial groove and, an often visible, cratered surface are always associated with very small and uniformly shaped B-Type granules. Type A granules appear damaged in few EN individuals, which may be linked to enzymatic digestion (salivary amylase) although plant food processing could also result in starch damage based on experimental results (*Soto et al., 2019*; *Zupancich et al., 2019*).

### Subtype Ib

Significant unevenness in granule dimensions and shape was recorded in the EM and most of the LM individuals (e.g., PAD9 and VL45). Granules in this subtype are dimensionally variable and their shapes can range from round to oval (*Figure 4b*). The well-known limitations in the inclusion, preservation, and recovery of plant debris in dental calculus (*Radini et al., 2017*) might be responsible for not recognizing this subtype previously.

## Type II

Starch granules attributed to this type consist of large aggregates as well as clustered polyhedral/irregular granules (main axis ranging from 5 to 15 µm). They were retrieved from 10 individuals (2 EM, 11 LM, 2 M/N, and 3 EN) (*Table 2*; *Figure 4g*). The identification of archaeological specimens is based on published records (*Mariotti Lippi et al., 2015*) and our experimental reference (*Avena barbata* L., *A. strigosa* Schreb., and *A. fatua* L.) (Figure 7). Granules of this morphotype were grouped in the tribe Aveneae/Poeae based on the fact that such large aggregates are found mostly in the genus *Avena* L. (oat), which is very common in the region.

## Type III

Starch granules attributed to this type are characterized by a polyhedral to subpolyhedral 3D morphology, a central hilum, and fine cracks. They were recovered in 16 individuals (5 for EM, 16 for LM, 2 for M/N, and 1 for EN) (*Table 2*; *Figure 4c* and *Figure 5b, c*). These features are consistent with starch granules of the tribe Paniceae of the grass family Poaceae and very well known in ancient starch research (*Madella et al., 2013*). In our sample, starch granules assigned to type III reach 21 µm of maximum width, which falls within the size range found in several species of *Setaria* spp., *Panicum* spp., and *Echinochloa* spp. (*Lucarini et al., 2016*; *Figures 6 and 7*). Small granules characterized by a round to subpolyhedral 3D morphology, and a central open hilum have been attributed to the tribe Andropogoneae and are here described under the general group of 'millets' as is common practice (*Madella et al., 2013*).

## Type IV

Starch granules of this type are identified in 10 individuals (2 EM, 8 LM, 1 M/N, and 2 EN) and are recovered above 6 granules per specimen, with the exception of the only EN individual (*Table 2*). Diagnostic morphological characteristics for Type IV granules are known in ancient starch research and include a reniform shape in 3D, a collapsed/sunken hilum forming a deep fissure along almost the entire granule, and a size ranging between 12 and 35 µm (*Henry et al., 2011*). Small cracks were observed departing from such hilum and were very evident under cross-polarized light. In most cases the extinction cross was very bright and showed several lateral arms diverging from the hilum in correspondence with the cracks (*Figure 4h*). Moreover, lamellae were visible toward the outer part of the granules. All these features are very peculiar and diagnostic of starch granules included in the species of the plant family Fabaceae (*Henry et al., 2011*), which is mostly known for its several edible domesticated species of legumes (e.g., *Lens culinaris* Medikus, *Vicia faba* L., and *Pisum sativum* L.), but also has a number of wild edible such as vetches (*Vicia* spp.). While many edible species of the family Fabaceae grow in the Balkans (e.g., *Vicia sativa* L., *V. cracca* L., *V. hirsuta*, *V. ervilia*, *Lathyrus pratensis* L., and *L. sylvestris* L.), an identification at species or genus was not possible due to overlaps in shape and size of starch granules at tribe level, which were observed in our modern reference collection (*Figure 7*).

## Type V

Few starch granules attributed to this type have been identified in eight individuals (2 EM, 5LM, nd 1MN) (*Figures 4d and 5a*; *Table 2*). Starch granules reach 23 µm in length and are mostly triangular with round corners and/or have an irregular oval shape (*Figure 5a*). Overall, lamellae can rarely be visible. The granules show a linear fissure in the center and sometimes the hilum appears as a wide depression. Under polarized light, the hilum is mostly centric while the extinction cross has bent arms. This type was found to have a very close visual match with the starch found in acorns of oaks (*Quercus* spp.), a Fagaceae member and well known in ancient starch research (*Liu et al., 2015*; *Figure 7i, j*).

## Type VI

Granules ascribed to this type have been identified in two individuals (1 LM and 1 M/N) (*Table 2*). They are characterized by a round 3D morphology and a central hilum, which appears as a wide

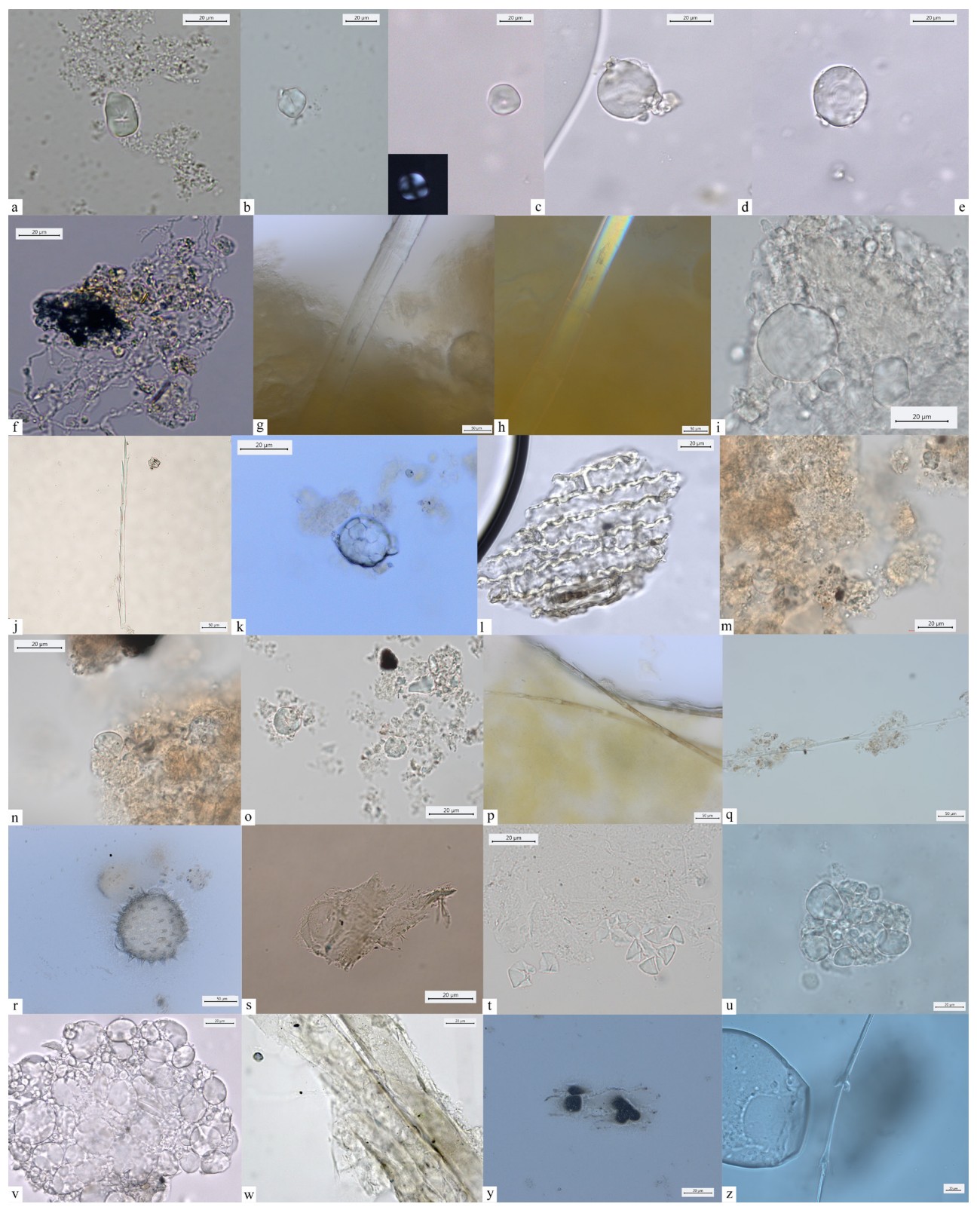

**Figure 5.** Other dietary and nondietary debris found in Mesolithic dental calculus from the Danube Gorges. *Early Mesolithic:* (**a**) Type V (PAD11); (**b, c**) Type III (PAD12); (**d**) Type I (A-Type granule) (PAD12); (**e**) Type I (A-Type granule) (PAD12); (**f**) smoke particle (LV50); (**g, h**) plant fiber embedded in calculus (PAD16); (**i**) Type Ib (PAD9); (**j**) feather barbule embedded in calculus (PAD9); *Late Mesolithic:* (**k**) Type II (PAD2); (**l**) polylobate phytolith (US64 x.11); (**m**) phytoliths (VL79); (**n–p**) Type VI (VL70,VL83); (**q**) feather barbules embedded in calculus (HV25/26); (**r**) echinate pollen grain in calculus (VL83); (**s**) plant

*Figure 5 continued on next page*

*Figure 5 continued*

tissue (LV79a); (**t**) Type II (VL43); (**u**) Type I (HV11); (**v**) Type Ia (HV11); *Mesolithic-Neolithic*: (**w**) Type I (HV16); (**x**) wood particle (PAD4); (**y**) phytoliths (LV28); (**z**) feather barbule (PAD4).

depression, and no lamellae or facets (*Figures 4i and 5n–p*). *Zarrillo and Kooyman, 2017* consider these morphological features diagnostic of some species of drupes and berries. In our sample, starch granules of this morphotype can reach 12 μm of maximum width, which is beyond species of berries and drupes in the Rosaceae family known in literature (*Zarrillo and Kooyman, 2017*) and in our modern reference (e.g., *Prunus spinosa* L.). Based on our experimental record, we assign type VI to species of the family Cornaceae (e.g., *C. mas*) (*Figure 7*), the remains of which are documented at Vlasac (*Filipović et al., 2010*).

In addition to starch granules, 42 phytoliths were retrieved in 24 individuals (EM = 4; LM = 13; MN = 2; N = 5). Mostly, short cells, commonly produced in leaves, stems and inflorescences, were identified and attributed to Pooid grasses. Multicellular structures of long cells were identified in Mesolithic individuals (HV25/26, VL70, VL79, and U222) (*Figure 4j and k*; *Figure 5m*). Of particular relevance is the recovery of multicellular phytoliths with dendritic appearance. It was observed at least in one case, still embedded in the dental calculus (*Figure 4k*). Single or multicellular dendritic structures were also identified in Neolithic individuals (VEL-2D and Gârleşti) (*Figure 4w, x*). A single polylobate cell was found in one LM individual (U64 x.11). With the exception of dendritic structures, characterizing grass

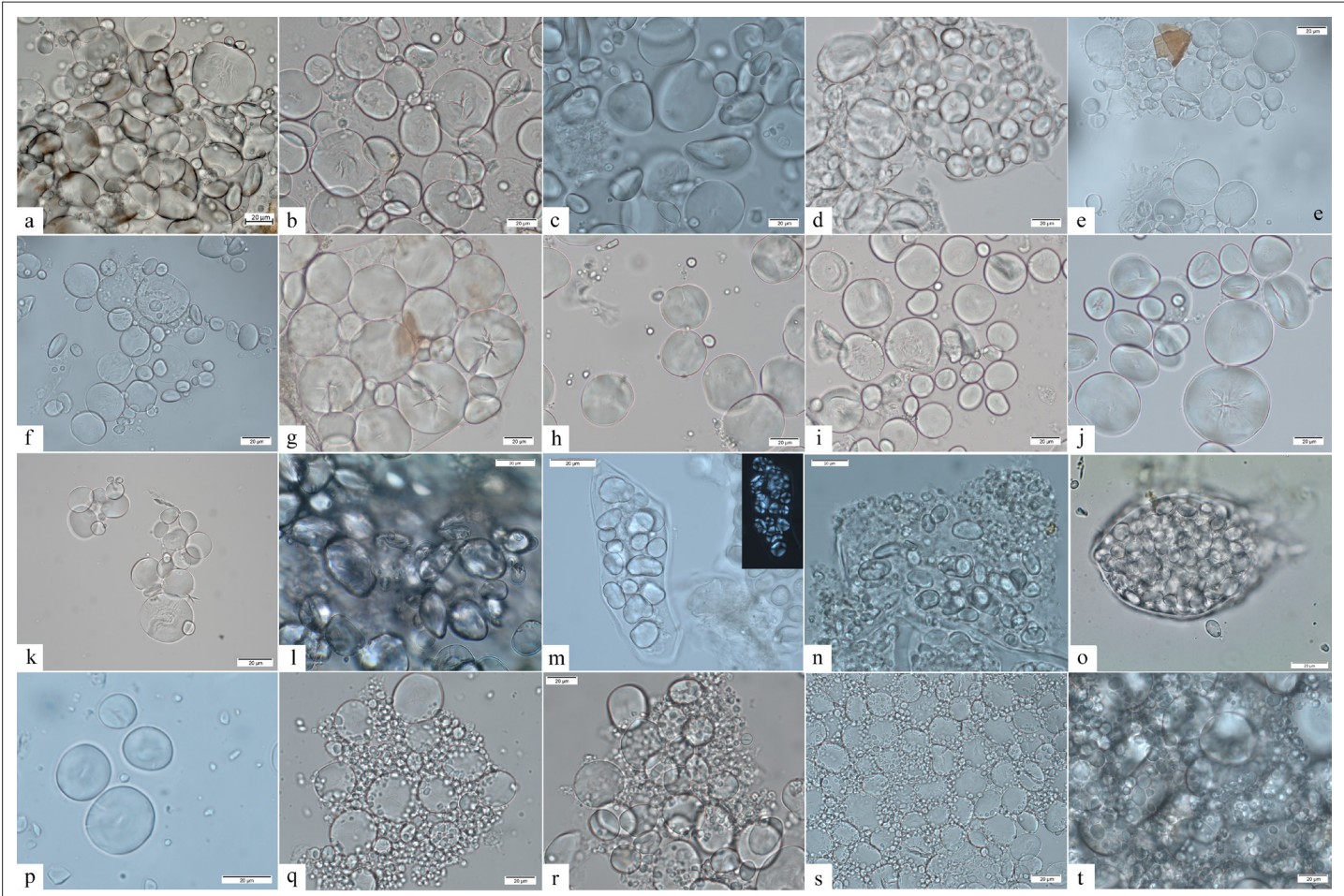

**Figure 6.** Starch granule morphological variability within the species of the genus *Aegilops* and domestic species of the Triticeae tribe. (**a**) *Aegilops cylindrica*; (**b**) *A. neglecta*; (**c**) *A. speltoides tauschii*; (**d**) *A. caudata*; (**e**) *A. triuncialis*; (**f**) *A. comosa*; (**g**) *A. uniaristata*; (**h**) *A. ventricosa*; (**i**) *A. geniculata*; (**j**) *A. crassa*; (**k**) *A. peregrina*; (**l**) *Elymus caninus*; (**m**) *Bromus tectorum*; (**n**) *Agropyron pungens*; (**o**) *A. farctus*; (**p**) *Dasypyron villosum*; (**q**) *Triticum monococcum*; (**r**) *Hordeum vulgare*; (**s**) *T. dicoccum*; (**t**) *T. aestivum*.

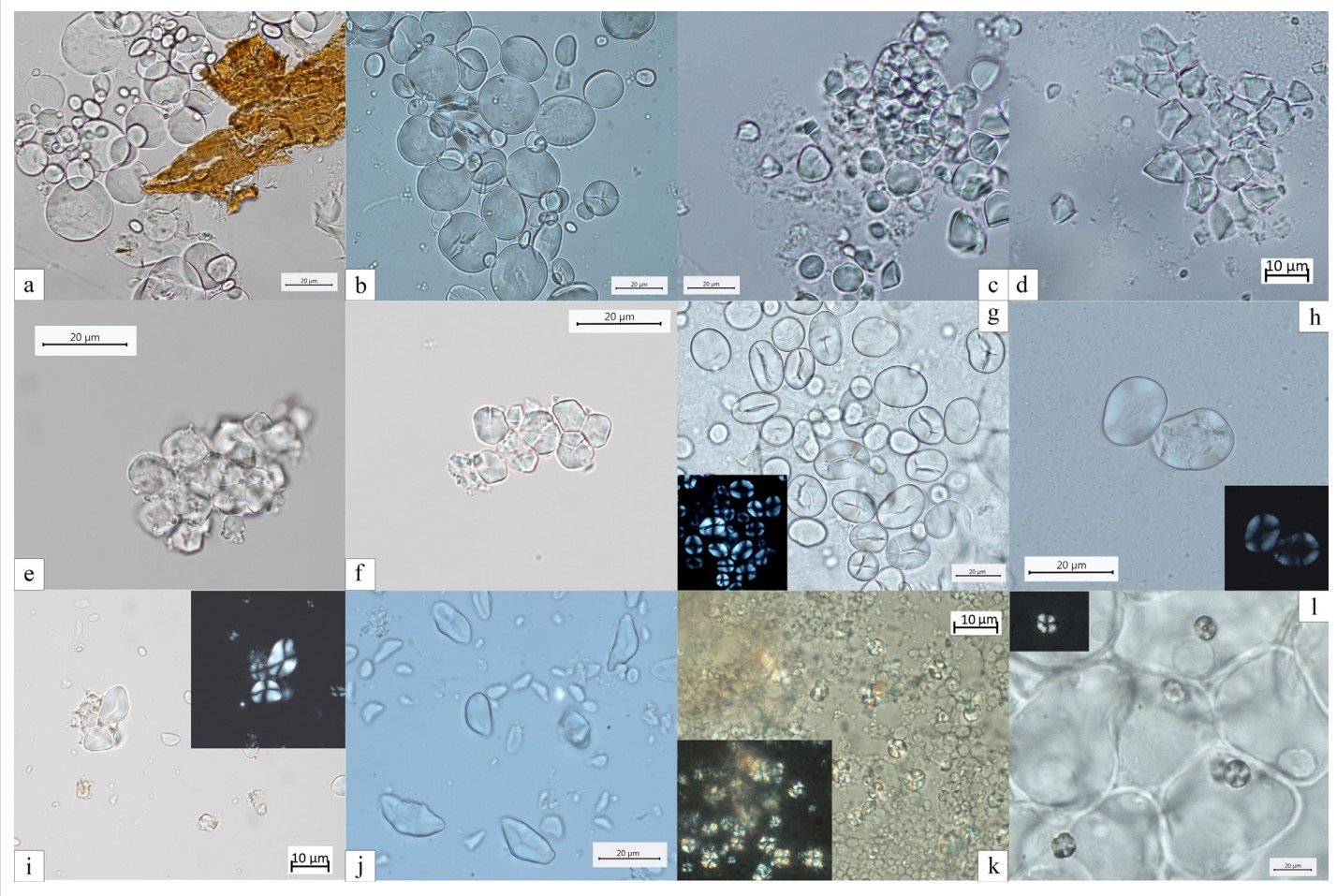

**Figure 7.** Experimental reference for starch granules identified in the dental calculus and ground stone tools. (**a**) *Aegilops triuncialis*; (**b**) *A. crassa*; (**c, d**) *Avena strigosa*; (**e, f**) *Setaria italica*; (**g**) *Vicia cracca*; (**h**) *V. sylvatica*; (**i**) *Quercus pubescens*; (**j**) *Q. robur*; (**k**) *Q. colurna*; (**l**) *Cornus mas*.

inflorescences, different nondietary reasons may be suggested for the inclusion of phytoliths in dental calculus (i.e., accidental ingestion, inhalation, dust in the environment generated by the use of grasses in a variety of activities and uses, such as flooring and kindling) (*Norström et al., 2019*).

## Ground stone tools

Diagnostic use-wear and residues are identified on 44 GSTs from the site of Vlasac. Analyzed tools included functional categories such as handstones (e.g., grinders and crushers) as well as passive bases (anvils) (Table 3). All of the tools are made of sandstone, characterized grains ranging in size between 0.2 and 1 mm densely distributed within the matrix. The combination of different functional modifications (i.e., flattened surfaces, pitted areas, rounding, etc.) on the single specimens suggests the long and complex life histories of the artifacts, often used in different activities. Within the tools displaying diagnostic use-wear, a total of 17 GSTs have surfaces bearing functional areas positively associated with plant food processing (*Table 3*; *Figure 3*). The analysis conducted at low magnification on these tools revealed macrotraces resulting in leveled surface crystal grains, sometimes covered by spots of yellowish organic film (sometimes striated) and white compacted powder (*Figure 4Z, Aa*). At a high magnification, high and low microtopographies of the GST surfaces are affected by a smooth domed, and sometimes striated, micropolish (*Figure 4Bb, Cc*). The aforementioned combination of use-wear and macroresidues are commonly associated with GSTs used as handstones for crushing and grinding grass grains and/or fruits, such as hazelnuts and/or acorns in our experimental record (*Cristiani and Zupancich, 2020*; *Figure 8*).

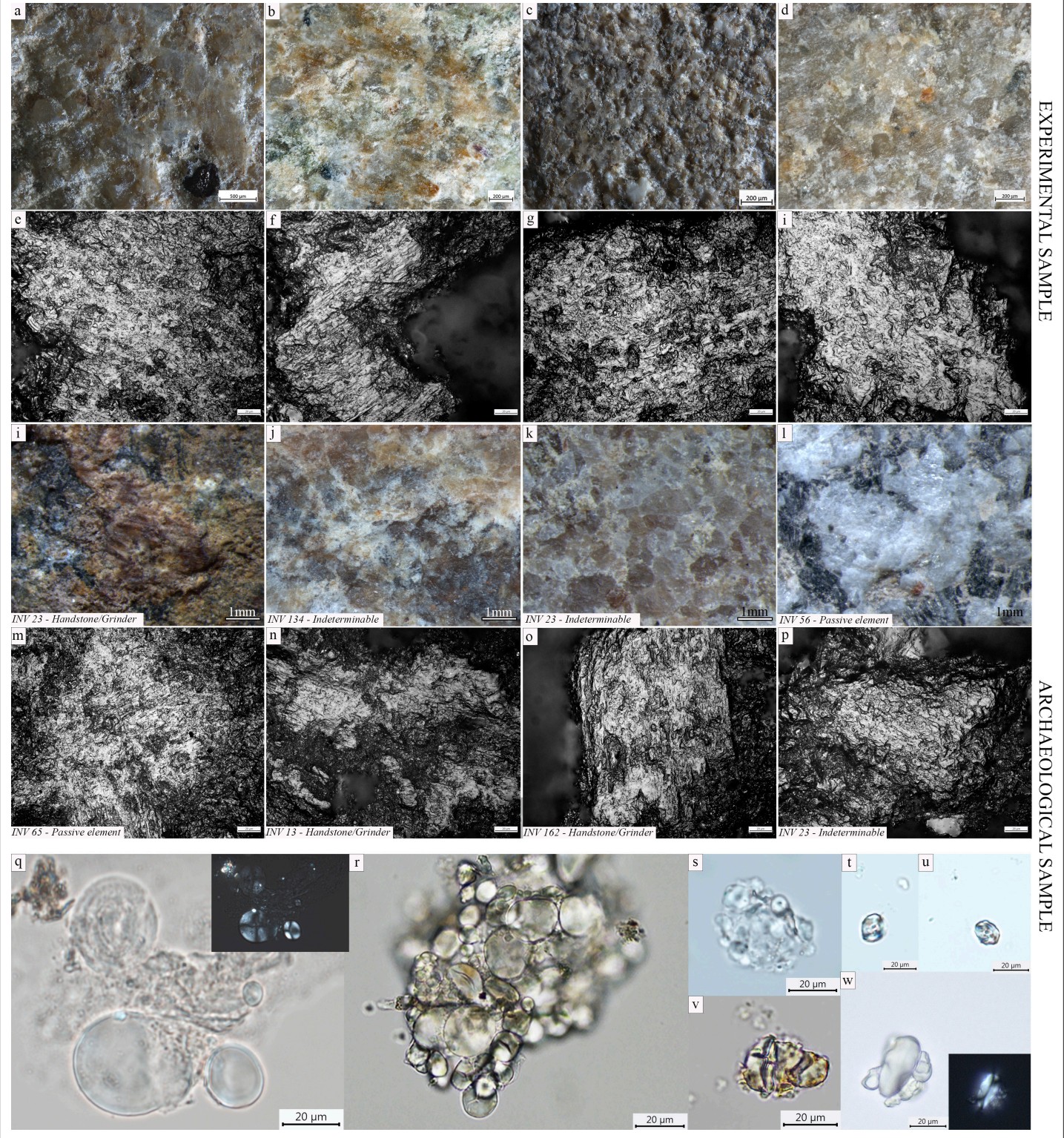

**Figure 8.** Experimental macroresidues and micropolish associated with grass grains processing compared to macroresidue and micropolish identified on archaeological ground stone tools from the site of Vlasac. (**a–e**) Yellowish organic film covering the crystal grains on experimental GSTs used to process oat (**a**), downy brome (**b**), wild grass grains (**c**), and millet (**d**); smooth domed and flat micropolish developing over the high and low microtopographies associated with oat (*Avena barbata*) grinding; (**f**) smooth flat and domed micropolish developing over the surface high and low microtopographies and characterized by narrow microstriations associated with grinding downy brome (*Bromus tectorum* L.); (**g**) smooth flat micropolish developed over the high and low microtopographies characterized by sporadic narrow striations associated with grinding wild grass grains (*Aegilops*

*Figure 8 continued on next page*

Figure 8 continued

*ventricosa* Tausch); (**h**) smooth domed polish developed over the high and low microtopographies associated with the grinding of foxtail millet (*Setaria italica* (L.) P. Beauvois); (**i–l**) spots of organic film, yellowish in color covering the crystal grains across the surface of archaeological GSTs; smooth domed micropolish identified on archaeological GSTs developing over the high and low surface microtopographies and associated with microstriations (**m-o**). Starch granules identified on archaeological GSTs. (**q**) Type I (GST no. INV.80); (**r**) Type I (GST no. INV.146); (**s**) Type III (GST no. INV.28); (**t**) Type VI (GST no. INV.67); (**u**) Type VI (GST no. INV.10); (**v**) Type VI (GST no. INV.146); (**w**) Type I (GST no. INV.71).

A total of 137 starch granules have been retrieved from the surfaces of the GSTs characterized by plant-related functional microscopic features. The optical and morphological properties of the starch granules support their attribution to morphotypes already documented in dental calculus from the Danube Gorges sites: Type I assigned to caryopses of the tribe Triticeae (76) (*Figure 4Dd, Ee, Ff*); Type IV, assigned to the Fabaceae family (13) (*Figure 4Ii*); Type III assigned to the tribe Paniceae of the grass family Poaceae (16) (*Figure 4Gg*); and type VI, assigned to berries of the family Cornaceae (32) (*Figure 4Hh, Jj*).

In sum, several hundreds of starch granules and phytoliths of grass grains of the Triticeae tribe have been identified in the analyzed dental calculus of the Mesolithic population in the Danube Gorges. In addition, residues and use-wear identified on GSTs from LM Vlasac show the existence of a plant food processing technology during this period aimed at preparing a coarse-grained flour through a combination of pounding and grinding gestures (*Table 3*; *Figure 3* and *Figure 9*). Grit particles, often retrieved in the analyzed dental deposit (*Table 2*), further confirm the use of sandstone GSTs in food processing. The conclusion about the consumption of partially processed grains is corroborated by the presence of starch granules still lodged in their amyloplast on Mesolithic GSTs and dental calculus, as suggested in our previous study (*Cristiani et al., 2016*). Interestingly, A-Type granules in EN dental calculus are generally retrieved singularly and exhibit a damage pattern observed when producing fine-grained flour experimentally only through prolonged bidirectional grinding (*Dietrich et al., 2019*). The pattern of bidirectional grinding is not documented on the examined LM GST from Vlasac, suggesting the existence of two different grain processing modalities typical of respective Mesolithic and Neolithic cultural traditions.

## Discussion

For some time now, there has been a recognition of the importance of plant foods in forager diets, based on both archaeological and ethnographic evidence (*Clarke, 1978*; *Lee et al., 1968*). Research on mineralized dental plaque has significantly advanced our awareness about ancient preagrarian food choices in Europe, Asia, and Africa, thanks to the dental plaque's potential to preserve plant microremains (*Buckley et al., 2014*; *Cristiani et al., 2018*; *Cristiani et al., 2016*; *Cummings et al., 2016*; *Nava et al., 2021*; *Norström et al., 2019*). However, in many archaeological case studies dating to early prehistory, preservation or recovery biases render plant food evidence invisible. In exceptional cases, good preservation has allowed for the remains of plant macroremains to be found, such as wild cereals at the Epipaleolithic site of Ohalo II in Israel, dating to 23 kya (*Nadel et al., 2015*; *Piperno et al., 2004*), or parenchyma remains at the Gravettian site of Dolní Věstonice in the Czech Republic (*Pryor et al., 2013*). On the other hand, microremains of oat caryopses have been detected on GST found at the Gravettian site of Paglicci cave in Italy (*Mariotti Lippi et al., 2015*). In a seminal synthesis about plant foods in the European Mesolithic, *Zvelebil, 2014* reviews macro- and microbotanical, palynological, artifactual (antler hoes, mattocks, GST), and human osteological (dental size and presence of caries) evidence for the consumption of nuts and fruits by European Holocene foragers, arguing for a form of niche constructing in temperate woodlands by means of deliberate forest clearance in order 'to increase the productivity of nut and fruit trees and shrubs, wetland plants, and possibly native grasses' (*Zvelebil, 2014*). The emphasis is on the existence of some form of husbandry of wild plant species, which did not necessarily lead to domestication. Furthermore, between 200 and 450 indigenous European edible plants (grass seeds, nuts, fruits, roots, tubers, and pulses) are found concentrated in wetland (coastal, lacustrine, and riparian) habitats (*Clarke, 1978*). Despite preservation and recovery problems, hundreds of Mesolithic sites across Europe have yielded the remains of hazelnuts, acorns, water-chestnuts, and other remains (*Zvelebil, 2014*).

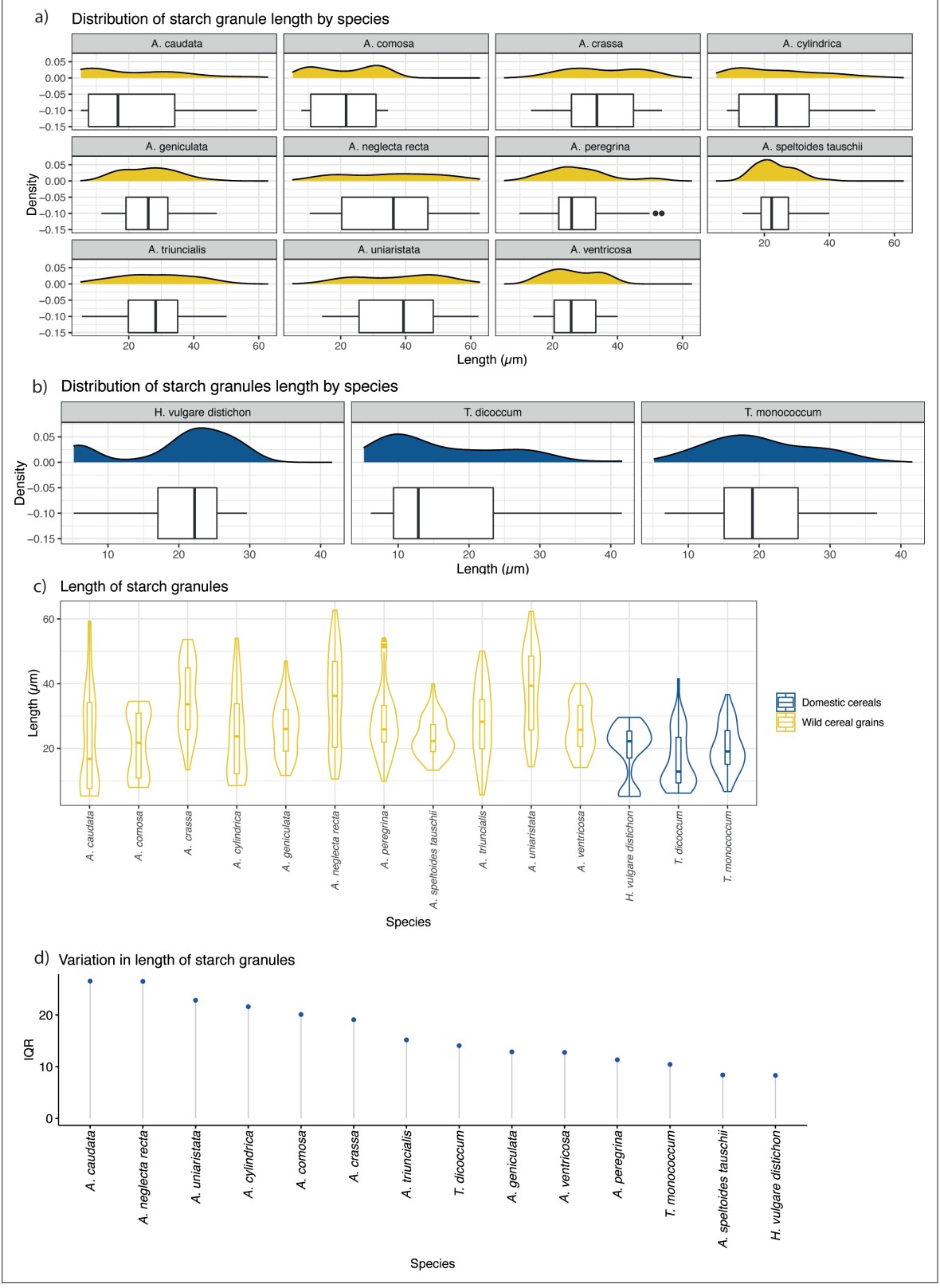

**Figure 9.** Starch granule length in modern wild and domestic cereal species. (**a**) Distribution of starch granule length in wild species; (**b**) distribution of domesticated species; (**c**) violin plot of comparing the length of starch granules in wild and domesticated species; (**d**) interquartile ranges (IQRs) of wild and domestic species. IQR corresponds to the difference in the medians of the lower and upper half of the data.

*Figure 9 continued on next page*

*Figure 9 continued*

The online version of this article includes the following figure supplement(s) for figure 9:

**Source data 1.** Starch granule length in modern wild and domestic cereal species.

In this paper, two complementary lines of evidence that we examined provide the first unambiguous and direct support for the consumption and processing of Poaceae grains among other types of edible plants by the Early Holocene foragers in the Danube Gorges area. The chronological framework of the analyzed sample suggests that this interest in and familiarity with various species of wild grasses of the Triticeae tribe (namely grass grains of the genus *Aegilops*) dates back to at least ~9500 cal BC. Macrobotanical remains belonging to this genus have not been recovered in Mesolithic and EN sites in the central Balkans (*Cristiani et al., 2016*: 10301). However, such absence in the archaeobotanical record could be the result of a host of taphonomic and recovery problems and should not be used to exclude the use of this genus during the Mesolithic (contra *Cristiani et al., 2016*: 4). Newly obtained evidence from the analysis of 61 individuals, which now also involves several EM individuals, lead us to suggest that *Aegilops* species were consumed in the region since the beginning of the Holocene. The mentioned difficulties associated with the recovery and preservation of botanical remains in local early prehistoric forager contexts along with some voids in the extant data regarding plant use by Mesolithic groups underline the significance of our findings based on the application of relatively recent advances in dental calculus and GST analyses.

We have previously argued that three LM individuals from the site of Vlasac dated to the mid-seventh millennium cal BC (H53, 64.x11, and H232), as well as two presumed EN individuals from Lepenski Vir (8 and 20) exhibit starches consistent with domesticated cereal species, such as *Triticum monococcum* L. (einkorn wheat), *Triticum dicoccum* L. (emmer wheat), and/or *Hordeum distichon* L. (barley) (*Cristiani et al., 2016*). This observation was based on the bimodal pattern of starch granules distribution, commonly attributed to domestic species and absent in most of the wild species of the genus *Aegilops* (*Howard et al., 2011*; *Stoddard, 1999*). This bimodal pattern is now further retrieved in two other individuals from two different sites (H326 from Vlasac and HV16 from Hajdučka Vodenica) dating to the LM and thus predating the arrival of full-blown agriculture in the region. However, burial 20 from Lepenski Vir, previously published as dating to the EN, has recently been directly AMS-dated to the EM (*Table 1*). This new chronological attribution does not correspond with our expectations that domesticated grains were introduced in the Danube Gorges area only in the LM. At the face of the current evidence, we explain this inconsistency in our results by arguing that the admittedly small number of starch granules found in this archaeological specimen might have affected the visibility of the potential variation in A-Type *vs.* B-Type population. Moreover, a large variability of starch granule distributions among different species of the Triticeae tribe has been acknowledged in the literature (*Henry and Piperno, 2008*; *Yang and Perry, 2013*) and supported by our experimental reference (*Figure 9*). Furthermore, fluctuations in environmental and growing conditions have also been recognised as relevant factors affecting starch granule size distribution (*Stoddard, 1999*; *Stoddard and Sarker, 2000*).

In addition to starch granules, other microremains, such as phytoliths and burnt debris, were recovered in the dental calculi of the analyzed Mesolithic population (*Figure 5l, m, y*). The paucity of archaeological phytoliths calls for caution when interpreting their dietary origin. Yet, the presence of few dendritic phytoliths in local forager dental calculus is likely related to plant consumption, as such microremains have experimentally been associated with mechanical destruction of husks and culms of Pooids by grinding (*Portillo and Albert, 2014*). In the investigated population, phytoliths could potentially provide means of understanding the use of plants as kindling and exposure to potential respiratory irritants generated during daily life activities, but their pathways are too many to narrow them down, and further work is required to better understand the origin of burnt material in dental calculus (*Radini et al., 2019*; *Radini et al., 2017*; *Scott and Damblon, 2010*). The retrieval of phytoliths from herbaceous plants that appear burnt could suggest that plant materials, potentially harvested, were used in a diverse range of activities, including kindling and roofing. Phytoliths of such plants might have also been naturally present in the environment, water, and soil (*Norström et al., 2019*). Such burnt debris may have reached the mouth by accidental ingestion through food and/or breathing. Other microdebris, such as nettle and wood fibers, might have also been linked to textile.

Pathways to wood inclusion in calculus vary from the use of a toothpick to crafting activities (*Radini et al., 2016*) and the production and maintenance of weapons, such as arrowheads.

For the moment, it remains unclear to what extent plant foods, and species of the Poaceae family, contributed to the diet of the Danube Gorges foragers, and it is equally difficult to be more specific about the range of activities involved in their acquisition prior to processing and consumption – from sporadic collecting of wild grasses to forest clearances, or some form of indigenous system of plant management. Yet, it seems clear from our data that this specific subsistence practice was passed over generations in this regional context up to the first contacts of these foragers with Neolithic, farming groups in the second half of the seventh millennium BC. Previously, we suggested that some sort of exchange between LM foragers and first Neolithic groups in the southern Balkans might have allowed for the introduction of the domesticated species of Triticeae in the Danube Gorges area ~6500 cal BC (*Cristiani et al., 2016*). Now, our extended study seems to suggest that this introduction of domesticated cereals, that is more productive cereal strains, from southwestern Asia was preceded by several millennia of collecting and consumption of native grass grains. Adoption of previously unknown plant foods is often facilitated by those local foods that in shape and taste resemble the new arrivals (*Livarda, 2010*). Incipient management practices on certain wild plant taxa have been documented among nonagricultural societies for environmental and/or cultural reasons, for protecting or promoting the relative abundance of a species, or for reducing the energy involved in its harvesting (*Fuller et al., 2011*; *Smith, 2012*).

Another recent study of dental calculus from the Danube Gorges area has reached a conclusion that domesticated cereals started to be used in this region only with the start of the Neolithic (*Jovanović et al., 2021b*). In the same study, the individuals dated to the EM exhibit a high occurrence of starch granules, which we have shown to be compatible with a variety of wild species of the Triticeae tribe (e.g., genus *Aegilops*). *Jovanović et al., 2021b* disregard the evidence of *Aegilops* consumption during the EM as an 'implausible pattern'. However, our results based on the combination of a robust sample of analyzed individuals, GSTs, and experimental data are in opposition to the conclusions of the mentioned study.

Most of prehistoric forager groups might have had regular access to plant nutrients and some sort of dependency on specific plants (*Fuller et al., 2011*). Accordingly, in our analysis of dental calculus and GSTs, we have shown that besides wild grasses, local foragers consumed oat, legumes, minor millet species of the genus *Echinochloa* and/or *Setaria* known as 'forgotten millets' (*Madella et al., 2013*; *Weber and Fuller, 2008*), acorns, and Cornelian cherries (*Figure 4*; *Table 1* and *Table 2*). Even if secure identification to species or genus level was not possible in every case (*Soto et al., 2019*), the morphological characteristics of starch granules systematically encountered in our archaeological samples, and the data available for the analysis of material culture, clearly show a contribution to the diet from these plant taxa.

Several starch granules of the tribe Aveneae have been identified in dental calculus, especially during the LM. Very abundant in temperate ecosystems, oat caryopses have been processed as foodstuff already during the Upper Palaeolithic at Paglicci Cave, in the south of Italy (*Mariotti Lippi et al., 2015*). They have also been documented in the dental calculus of the aforementioned LM individual from Vlakno Cave in the Eastern Adriatic region (*Cristiani et al., 2018*). Overall, starch granules of this tribe were remarkably well preserved in our dental calculus samples and large aggregates characterizing Aveneae were abundant. Interestingly, starches of this tribe recorded on archaeological GSTs, and associated with the production of flour, consist of abundant single sparse polyhedral granules (*Mariotti Lippi et al., 2015*). In our case study, exceptionally well-preserved aggregates of Aveneae have been identified in dental calculus only. The concentration of semiopen aggregates (*Figures 4g, 5k and u*) sustains the hypothesis of a coarse processing of the seeds during the Mesolithic, already suggested for other grass grain taxa.

Small starch granules from grasses of the Poaceae family have been attributed to the Andropogoneae and Paniceae tribes (*Figure 5B, C*; *Madella et al., 2013*). Our results confirm previous conclusions about the use of grasses of the tribe Paniceae by the LM foragers (*Cristiani et al., 2016*), while extending the consumption of this general types of millets to EM foragers too. However, no evidence of this food has been ascertained on the basis of stable isotope analysis (*Borić et al., 2004*). This pattern could mean that their consumption was not predominant in dietary practices of these Mesolithic foragers. We stress that a great variety of species of $C_4$ plants belonging to the tribe

Paniceae and the tribe Andropogoneae (e.g., species of plants of the genera *Setaria* Beauv. and *Echinochloa* P. Beauv.) grow well nearby water environments and slow-flowing waters, and it is very likely that a mixture of species from such genera might have contributed to the Mesolithic diet. Furthermore, they often grow in association with each other. The presence of Paniceae and the Andropogoneae tribes, combined with the evidence of several feather barbule fragments from aquatic birds, clearly point to a familiarity with resources from riverine environments other than fish.

Species of the family Fabaceae are well represented in the EM and LM individuals from the Danube Gorges area. While a secure identification to species or genus was not possible in our samples, we know that wild pulses (*Lens* sp. Mill.) and bitter vetch (*V. ervilia* (L.) Willd.) were used as foodstuffs for the Mesolithic inhabitants of Franchthi Cave (*Asouti et al., 2018*; *Van Andel et al., 1987*). Further evidence for a pre-Neolithic consumption of the species of the tribe Fabeae comes from Uzzo Cave, Italy, where wild legumes (*Lathyrus* sp. L., *Pisum* sp. L.) are well represented along with other arboreal fruits (*Arbutus unedo* L.), acorns (*Quercus* sp. L.), and wild grapes (*Vitis vinifera* subsp. *sylvestris* L.) (*Costantini, 1989*), as well as from the site of Barma Abeurador in southern France (*Vaquer et al., 1986*). In addition to the evidence from dental calculus, indirect evidence for the consumption of species of the family Fabaceae has also been retrieved from one GST (*Figure 4li*).

Species of the Fagaceae family (cfr. *Quercus ilex* L., *Quercus* sp. L.) were also a significant food resource for the Danube Gorges foragers (*Figure 5A*). Fragments of acorns were found at several Mesolithic sites across Europe (*Holden et al., 1995*; *Kubiak-Martens, 1999*) and in the investigated region (*Mason, 1995*).

Evidence for the consumption of Cornelian cherries (*C. mas* L.) has been retrieved in dental calculus belonging to LM and Mesolithic-Neolithic individuals from Vlasac (*Figure 5n–p*) and also on eight GSTs from the same site. The results based on the analyses of GSTs not only corroborate the evidence for the use of these fruits derived from the calculus analysis, but also support the role of such plants in Mesolithic daily life. Cornelian cherries are the most recurrent macroremains at Vlasac and are documented across Europe as a Mesolithic source of food and medicine (*Divišová and Šída, 2015*; *Filipović et al., 2010*). Ethnographically, various fruits commonly referred to as 'cherries' and/or 'berries' are particularly well documented among Native American groups (*Siegfried, 1994*; *Zarrillo and Kooyman, 2017*). Throughout much of North America several species of the Cornaceae and Rosaceae plant families were processed using unmodified pounding stone tools to add to pemmican or to dry as cake. Interestingly, Woodland Cree people used to mix berries and cherries with fish eggs (*Siegfried, 1994*) and meat (*Lowie, 1922*).

We conclude that the long-lasting interactions with edible grains (but also wild pulses), documented in the Balkans since the end of the Pleistocene might have allowed enough time for specific eating habits, tastes, and 'cultural valuation' (*Hastorf and Foxhall, 2017*) to develop. Such a shared knowledge about specific plant resources effectively predates the introduction of agriculture in Europe, and might have eventually eased the introduction of domesticated species starting from the second half of the seventh millennium BC. Our results call for more systematic and interdisciplinary research in order to reconstruct plant food traditions and cultural tastes before the introduction of agriculture.

## Materials and methods

The examined collection of the teeth previously studied for strontium isotopes (*Borić and Price, 2013*) from five sites in the Danube Gorges area and the site of Gârleşti from the region of Oltenia in Romania and additionally collected teeth from the sites of the Velesnica and the 2006–2009 excavations at Vlasac contained a total of 155 specimens. Of these, a total of 60 individuals had sufficiently preserved calculi for further analyses (*Table 1*): 13 individuals date to the EM, 29 to the LM, 9 to the Mesolithic–Neolithic transition phase (M/N), and 8 to the EN. In addition, two later period burials were also included in the analysis as a methodological comparison (*Table 1*). In order to corroborate data obtained through dental calculus analysis, we also analyzed 101 sandstone GSTs of the 131 implements found at the site of Vlasac during the 1970–1971 excavations and chronologically attributed to the LM (*Borić et al., 2014*; *Srejović and Letica, 1978*). While GSTs are well documented during the Mesolithic–Neolithic transition at the site of Lepenski Vir (*Antonović, 2006*), earlier evidence for their use is available only from the site of Vlasac. This site yielded the richest Mesolithic assemblage of nonflaked stone tools recovered so far in the region (*Antonović, 2006*; *Borić et al., 2014*; *Srejović*

*and Letica, 1978*). GSTs underwent a functional study aimed at verifying their function and potential involvement in plant food processing (*Table 3*; *Figure 3*).

## Dental calculus – sampling, extraction, decontamination, and examination procedures

All the sampling was conducted under the stereoscope, as can be seen in *Figure 2* (for more details see below), and following strict protocols systematized by Sabil and Fellow Yates (*Sabin and Fellow Yates, 2020*) with some variation (disposable blades were changed after each sample extraction). Deposits of dental calculus were judiciously left on the teeth for future research. Whenever possible, sampled dental calculus was further subdivided for metagenomic analysis aimed at reconstructing aDNA of oral bacteria (*Ottoni et al., 2021*).

Decontamination and the extraction procedures for microdebris were conducted in dedicated clean spaces not connected to modern botanical work and under strict environmental monitoring of the DANTE laboratory of Sapienza University of Rome (IT), the BioArch laboratory at University of York (UK), and the aDNA facility of the University of Vienna (Austria). In all of these facilities, strict contamination rules were followed. Cleaning is carried out daily and no food is allowed in order to prevent any type of modern contamination. Bench space surfaces were cleaned prior to the analysis of each sample, using soap and ethanol, followed by covering of the surfaces by aluminum foil, and using of clean starch-free nitrile gloves at all times. Calculus cleaning was carried out under the stereomicroscope, on a Petri dish previously washed and immersed in hot ultrapure water, with magnifications up to ×100. The removal of the mineralized soil adhering on the surface of the calculus was meticulously carried out using sterile tweezers to hold the sample and a fine acupuncture needle to gently scrape off the soil attached to the external layer of the mineralized plaque. The procedure was performed using drops of 0.05 M hydrochloric (HCl) acid to dissolve the mineralized flecks of soil and ultrapure water to block the demineralization, as well as to wash and remove the contaminants. Once the calculus surface was cleaned, the contaminated soil was checked for possible cross-contamination and the clean samples were washed in ultrapure water up to three times in order to remove any trace of sediment. The clean calculus was then dissolved in a solution of 0.5 M HCl and subsequently mounted on slides using a solution of 50:50 glycerol and ultrapure water. Furthermore, control samples from the clean working tables and dust traps were collected and analyzed for comparative purposes in order to prevent any type of modern contamination in these laboratories – this is a practice routinely done in our laboratories, even at times where no archaeological analysis occur, to allow a better understanding of the flow of contaminations through seasons. Our results based on this procedure show that synthetic and plant fibers and hairs, fungal spores and hyphae, palm and conifer pollens, insects' debris; maize starch granules were detected twice while unidentified small starch granules were very rare; phytoliths and starch granules belonging to species of the Triticeae tribe were never recovered (*Figure 10*). We did not retrieve any debris morphologically similar to any of the remains in the environmental control samples. Furthermore, starch granules amounted to a neglectable fraction of the laboratory 'dust' – suggesting it is extremely unlikely that an event of contamination of starch granules would occur in the lab, where no other remains from dust, way more common, were not found.

The examination of microdebris embedded in the calculus matrix was performed at Sapienza University of Rome and at the University of York using a Zeiss Imager2 cross and an Olympus polarized microscope with magnifications ranging from ×100 to ×630. A modern reference collection of 300 plants native to the Balkans, the Mediterranean region, and Europe was used as a comparison, along with published literature, for the identification of archaeological starch granules. The experimental reference collection also included species documented in the local archaeological record (*Filipović et al., 2010*; *Marinova et al., 2013*).

## GSTs functional analysis

GSTs were sampled and analyzed at the Archaeological Collection of the University of Belgrade. Strict protocols were followed for controlling modern contamination during the residue sampling and functional analysis of GSTs: bench surfaces where the work was conducted were cleaned before the analysis of each tool using ethanol, hot water, and covered by aluminum foil; starch-free gloves were used while handling the GSTs; dust samples from the storage boxes and the working tables were collected and analyzed for comparative purposes; use-wear and residue analyses were performed

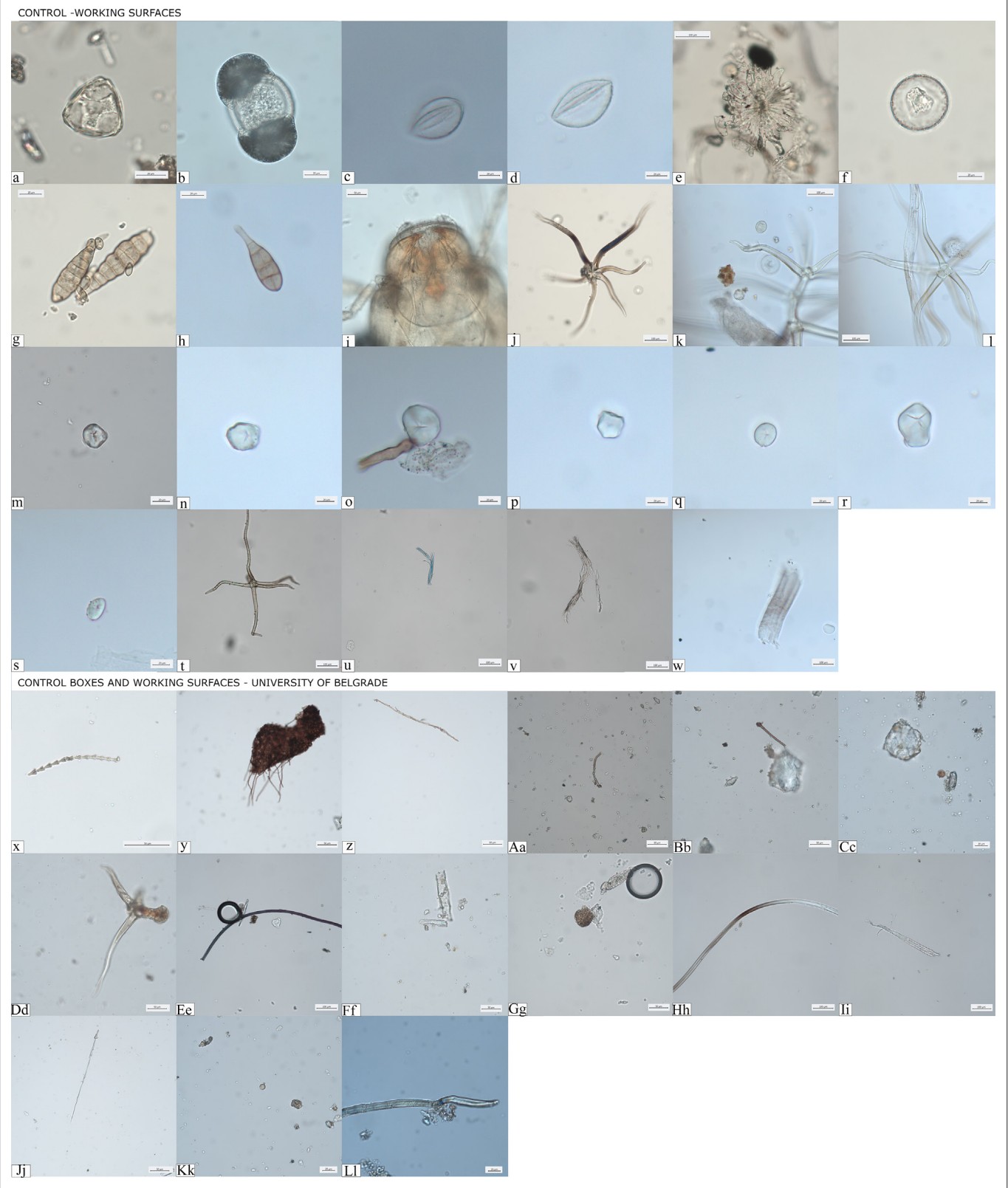

**Figure 10.** Controls for contamination. (**a–w**) Evidence of pollutants retrieved from clean working surfaces and dust traps located in different areas of the DANTE Laboratory at Sapienza University of Rome; (**x–Ll**) dust recorded in storage boxes where groundstone tools were stored at the Faculty of Philosophy, University of Belgrade.

on the surfaces not affected by severe postdepositional modifications and free from concretions. Furthermore, starch granules were considered reliable only when in combination with use-wear traces associated with plant processing.

Functional study involved the analysis of use-wear traces on the GST surfaces at low magnification (×0.75–×168) using a Zeiss Discovery V20 stereomicroscope and at high magnification (×200–×500) with a Zeiss AxioScope metallographic reflected light microscope(*Cristiani and Zupancich, 2020*; *Dubreuil et al., 2015*). At low magnification, GSTs were analyzed using a Zeiss Discovery V20 Stereomicroscope, which allowed us to assess the state of preservation of the materials and identify the residues still adhering to the surfaces. Appearance, morphological features, and spatial patterns of macroresidue distribution were considered (*Langejans, 2010*). Casts of the used areas were taken by means of a high-resolution polyvinylsiloxane (Provil Novo Light Fast Set), and later analyzed at high magnification (up to ×500) at the DANTE Laboratory at the Sapienza University of Rome. Micropolish, abrasions, and microstriations across the tools' surfaces were identified using a Zeiss AxioScope metallographic microscope, and described following relevant parameters available in literature (*Adams et al., 2006*; *Dubreuil et al., 2015*; *Hamon and Plisson, 2009*).

Microresidues were sampled before surface casts. Ultrapure water was placed on the crevices of surface and left for 1 min on the artifact in order to soften the residues, then pipetted out and stored in a sterile tube. Once in laboratory, the samples were centrifuged and the natant mounted on microscope slides using a 50% solution of purified water and glycerol. Slides were subsequently analyzed in transmitted light using Zeiss Imager2 microscope (×630) and cross-polarizing filters. For archaeological residues, appearance, morphological features, and spatial patterns of distribution were considered (*Cristiani and Zupancich, 2020*).

An experimental reference collection of used GSTs and starch granules housed at the DANTE laboratory was consulted along with relevant literature and scientific databases. High-resolution images of the identified use-wear and residue were taken at ×630 using a Zeiss Axiocam 305 high definition color camera. Risk of modern contamination from the storage and sampling environment was minimized following a strict cleaning procedure before and during the sampling/analysis.

## Testing morphological differences in experimental starch granules from *Aegilops*, *Hordeum*, and *Triticum* species

Methodologically, we were able to characterize the morphological variability of starch granules in ancient plant species consumed in the investigated area, hence complementing our previous work and its implications (*Cristiani et al., 2016*). Differences in the starch granules assigned of the tribe Triticeae in the analyzed individuals were identified on the basis of (1) the specific morphology, dimensions, and appearance of Type A and B granules; and (2) the proportion between A-Type and B granules. In particular, during the EM A-Type granules preserved in calculi are very large and round in shape with deep lamellae visible only in the granules' mesial part. Additionally, B-Type granules with different sizes and shape have been recorded in all of the EM individuals and most of LM individuals (*Figure 4*). This feature is absent in the EN individuals analyzed in this work, displaying only identical small round B-Type granules, while A-Type granules are large, round to oval/lenticular, with lamellae less pronounced in the mesial part of the grains and well visible craters on their surfaces (*Figure 4y*). We could not match these differences in our experimental record, which includes various *Aegilops* species as well as wild species within the genera *Hordeum*, *Elymus* L., *Agropyron* Gaertn., *Dasypyron* L. growing locally, and in the literature (see *Henry et al., 2011*). The abovementioned features have consistently been assigned to modern domestic Triticeae species (*Triticum* spp. and *Hordeum* spp.) (*Cristiani et al., 2016*; *Piperno et al., 2004*; *Yang and Perry, 2013*). Given the high variability recorded in the dimensions and distribution of starch granules within the modern, locally available, species of the genus *Aegilops* (*Figure 6a–j*), further statistical work was carried out in order to interpret starch granules assigned to the tribe Triticeae in Mesolithic-Neolithic transitional contexts.

Caryopses from 11 *Aegilops* species (*A. triuncialis*, *A. comosa*, *A. crassa*, *A. cylindrica*, *A. geniculata*, *A. neglecta*, *A. speltoides tauschii*, *A. peregrina*, *A. triuncialis*, *A. uniaristata*, and *A. ventricosa*), 1 *Hordeum* species (*Hordeum vulgare distichon*), and 2 *Triticum* species (*Triticum monococcum* and *T. dicoccum*) grown in the central Balkans were collected. All plant material was grounded using pestle and mortar. Starch powder (0.5 mg) was resuspended in 100 µl of sterile distilled water and vortexed for 5 min. After that, the sample was observed by an optic transmitted light microscopy (*Figure 6*).

Fifty starch granules were randomly selected (for size and shape), counted, and their length measured. Minimum and maximum lengths, mean, and median values with relative standard deviations and inter-quartile ranges are reported for each species in *Table 4*. Length distribution and variation of modern starches are reported in *Figure 9*. Finally, length distribution of the starches from each species was compared with the other species to investigate the existence of significant differences. This statistical analysis was carried out through a pairwise Wilcoxon test (*Table 5*). Results were considered significant for p values <0.05 (<0.05; *<0.01; ***<0.001) and not significant (ns) for measurements >0.05.

Triticeae starches are known to possess a bimodal distribution, made up of small (B-Type) and large (A-Type) granules (*Figure 6*). In the analyzed *Triticum* and *Hordeum* genera, B-Type grains are more abundant than A-Type, except for *H. secalinum* Schreb., which does not exhibit the large granules. Differently, in *Aegilops* genus, the size distribution of starches is characterized by two different trends. The first one, evidenced in *A. caudata* L., *A. cylindrica* Host, *A. comosa* Sm., and *A. speltoides tauschii* Coss., appears very similar to that of *Triticum* and *Hordeum* samples, although B-Type grains are less abundant than their counterparts in wheat and barley. On the other hand, the second cluster (*A. crassa* Boiss. ex Hohen., *A. geniculata* Roth, *A. neglecta* Req. ex Bertol., *A. speltoides tauschii* Tausch, *A. triuncialis* L., *A. uniaristata* Vis., and *A. ventricosa* Tausch) exhibits larger starches, determining a significant shift of the mean size toward intermediate lengths. In general, the present experimental analysis revealed that *Aegilops* sp. starch granules show a larger size distribution than *Hordeum* sp. and *Triticum* sp. (*Figures 6 and 9*). This evidence is also supported by the pairwise Wilcoxon test, which highlights that *Aegilops* spp. starch measurements are significantly different from those obtained for *Hordeum* sp. and *Triticum* sp. counts (*Table 4*).

## Acknowledgements

The authors wish to thank the late Živko Mikić (University of Belgrade), the late Borislav Jovanović (Serbian Academy of Sciences and Arts), Duško Šljivar (National Museum in Belgrade), and Jelena Rankov (National Museum, Belgrade) for permissions to sample the osteological collections from the Danube Gorges. For the purpose of open access, the authors have applied a CC BY public copyright license to any Author Accepted Manuscript version arising from this submission.

## Additional information

### Funding

| Funder | Grant reference number | Author |
|---|---|---|
| H2020 European Research Council | 639286 | Emanuela Cristiani |
| National Science Foundation | BCS-0235465 | T Douglas Price<br>Dušan Borić |
| NOMIS Stiftung | | Dušan Borić |
| Wellcome Trust | 209869/Z/17/Z | Anita Radini |
| British Academy | SG-42170 | Dušan Borić |
| British Academy | LRG-45589 | Dušan Borić |

The funders had no role in study design, data collection, and interpretation, or the decision to submit the work for publication.

### Author contributions

Emanuela Cristiani, Conceptualization, Data curation, Formal analysis, Funding acquisition, Investigation, Methodology, Project administration, Resources, Supervision, Validation, Visualization, Writing - original draft, Writing – review and editing; Anita Radini, Andrea Zupancich, Formal analysis, Investigation, Methodology, Validation, Visualization, Writing - original draft, Writing – review and editing; Angelo Gismondi, Alessia D'Agostino, Formal analysis, Writing - original draft, Writing – review and editing; Claudio Ottoni, Formal analysis, Writing – review and editing; Marialetizia Carra, Writing

– review and editing; Snežana Vukojičić, Mihai Constantinescu, Dragana Antonović, T Douglas Price, Resources, Writing – review and editing; Dušan Borić, Conceptualization, Data curation, Funding acquisition, Investigation, Project administration, Resources, Supervision, Validation, Visualization, Writing - original draft, Writing – review and editing

**Author ORCIDs**
Emanuela Cristiani  http://orcid.org/0000-0002-2748-9171
Anita Radini  http://orcid.org/0000-0002-2099-2639
Andrea Zupancich  http://orcid.org/0000-0002-9503-1234
Angelo Gismondi  http://orcid.org/0000-0002-9257-9667
Claudio Ottoni  http://orcid.org/0000-0001-8870-1589
Dušan Borić  http://orcid.org/0000-0003-0166-627X

**Decision letter and Author response**
Decision letter https://doi.org/10.7554/eLife.72976.sa1
Author response https://doi.org/10.7554/eLife.72976.sa2

## Additional files

### Supplementary files
• Transparent reporting form

### Data availability
All data generated or analysed during this study are included in the manuscript and supporting file.

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
