## [Editor Report]

By combining starch grains analysis from dental calculus and grinding implements, the authors demonstrate the consumption of a large variety of plants by Mesolithic foragers and Neolithic farmers in the Danube Gorges of the Balkans. The data and analyses advance debates on the intensification of plant selection prior to their strict domestication.

---

## [Decision Letter]

**Decision letter after peer review:**

[Editors’ note: the authors submitted for reconsideration following the decision after peer review. What follows is the decision letter after the first round of review.]

Thank you for submitting the paper "Cereal grain consumption among Early Holocene foragers of the Balkans predates the arrival of agriculture" for consideration by *eLife*. Your article has been reviewed by 3 peer reviewers, and the evaluation has been overseen by a Reviewing Editor and a Senior Editor. The reviewers have opted to remain anonymous.

We are sorry to say that, after consultation with the reviewers, we have decided that this work will not be considered further for publication by *eLife*. However, all reviewers had positive views about your work and paper. Yet the consultation consensus was that the extent of methodological revision and interpretive adjustments that would be required was more than what we felt could be accomplished with a reasonable revision timeframe and effort. Please see the below reviews. If you were to expansively address the comments, and based on feedback from the reviewers, I would be willing to consider a revised version of the manuscript as a new submission, with a high bar of expectations (and no guarantee of full review).

*Reviewer #1:*

This manuscript presents the results of microbotanical (starch grains and phytoliths) and chemical analyses in dental calculus from 61 Mesolithic and Early Neolithic individuals from several sites in the Danube Gorges of the central Balkans. Microbotanical evidence from 44 grinding stones from one of the studied sites is presented as additional evidence supporting the consumption of plants (wild grasses, pulses and fruits) before the arrival of southwest Asian domestic crops.

Archaeologists have long acknowledged the use of plants by hunter-gatherer societies worldwide prior to the advent of agriculture. In the Introduction, the authors claim that "forager knowledge and consistent use of wild cereals are still debated and poorly documented outside of the assumed centers of domestication in southwestern Asia" (l. 40-41) and that "the hypothesis of a systematic use of wild cereals (e.g., Aegilops spp.; Hordeum spp.) in this region [the Balkans] during the Mesolithic has not been verified until now" (l. 46-47). These claims seem to justify the novelty of this study. However, the beginning of the Discussion reviews evidence for wild grass use by hunter-gatherers in southeastern Europe, including one of the study sites, in the form of macro and microbotanical remains and pollen grains. The present study, thus, adds to this body of evidence (which should be presented in the Introduction, not the Discussion) in a specific region.

The robustness of this study lies in the high number of samples analysed. However, I do not believe that the analysed datasets are comparable. Dental calculus and grinding stones do provide independent evidence for the consumption of plant resources, but only when analysed from the same chronological periods and the same sites. In this study, dental calculus samples come from six sites dating to the Early Mesolithic, Late Mesolithic, the Mesolithic-Neolithic transition and the Early Neolithic, but samples from grinding stones come from a single site and a single occupational deposit (Late Mesolithic). Moreover, the analyses conducted in each set of samples are not consistent either-microbotanical and chemical analyses in dental calculus vs. microbotanical analyses in grinding stones-further precluding the comparability of the resulting datasets. In my opinion, these samples are not comparable and should not be presented as part of the same study.

Regarding the identification of microscopic plant remains in samples from dental calculus and grinding stones, I am puzzled by the identification of some Type I starch grains (Triticeae tribe) as belonging to the genus Aegilops. In a previous study in which several of the authors of the current paper participated (Cristiani et al. 2016), the identification of Triticeae starch grains as Aegilops was specifically disregarded because "Aegilops spp. are absent from assemblages with analyzed macrobotanical remains found at Mesolithic and Early Neolithic sites in the central Balkans" (Cristiani et al. 2016: 4). However, in the present study the authors seem comfortable assigning part of the encountered Triticeae starch grains to this genus without rectifying their previous claim.

Also in regards to the starch assemblage, there seem to be two categories of Type I starch grains, one identified as Aegilops and another as a domestic species within the Triticeae tribe. If the authors believe that this is the case, I recommend separating them into subtypes and treating them a separate taxa for the morphometric analysis.

Regarding the starch grain morphometric analysis, I am troubled by the fact that the length of Type A and Type B starch grains from modern and archaeological Triticeae spp. seems to have been analysed together, instead of analysing Type A and Type B grains separately. This makes it impossible to compare the results of this study to previously published data (e.g., Henry et al. 2011; Yang and Perry 2013) and potentially biases the morphometric data from archaeological starch grains due to the fact that small starch grains (in this case, Type B) are less likely to survive than large starch grains (Haslam 2004).

The phytolith assemblage is smaller and it is unclear whether the authors consider it as evidence for food (l. 265-275) or as 'non-dietary remains' (l. 294). In any case, Figure 4w is mentioned as evidence for 'multicellular dendritic structures' (l. 270), but I do not believe that the image shown in Figure 4w can be qualified as a phytolith. Evidence for sedges is also mentioned in l. 298 but not presented elsewhere.

Overall, I believe that this study presents evidence for the consumption of wild grasses and other plants during the Mesolithic and the transition to the Neolithic in the Danube Gorges of the Balkans, but I do not think that the evidence is properly presented.

I find several of the terms employed by the authors inaccurate or problematic. Perhaps the most paradigmatic is the term 'cereal', used in the title, the Abstract (l. 27) and throughout the manuscript. The term 'cereal' refers to a cultivated plant within the grass family (Poaceae). I do not believe that the authors aim at implying the existence of 'low level food production' (Smith 2001), often referred to as 'cultivation' (e.g., Bowles 2011), in the Mesolithic Danube Gorges, so I recommend the consistent use of the term 'wild cereal' (e.g., l. 33) or the more accurate 'wild grasses'. Other terms which I find potentially problematic include 'complex foragers' (e.g., l. 31)-what is a 'complex forager' and how does it compare to a 'simple' (?) forager? – and the use of the term 'microfossils' when referring to ancient starch grains, which are not fossilised and should therefore be referred to as 'microremains' or 'microscopic remains'. Finally, the term '"forgotten" millets' (l. 478) is used without explaining what is meant by 'forgotten'. If it refers to the fact that the millet species mentioned in the manuscript are no longer consumed in the Balkans, how is it different from Aegilops?

*Reviewer #2:*

This paper contributes significantly to the debate on the possible intensification of plants exploitation by late foraging populations prior to strict neolithisation. It focuses on a particular kind of botanical remains, starch grains, from two different kind of sources: dental calculus and ground stone tools. The region understudy, the Balkans, is particularly adapted to discuss this question, while as a series of important Mesolithic and Neolithic sites are well excavated in the considered area. The corpus of individuals is important, and allows solid interpretations. The results presented are convincing, and shows the high diversity of plants consumed by foragers. They also bring solid arguments to demonstrate that Late Mesolithic foragers were engaged in the consumption and probably selection of several of the wild plant species that were cultivated in their domestic form during the Neolithic. It keeps open some questions regarding the exact datation of the domestication, at the heart of intense debates among prehistorians.

A couple of improvements could be proposed to enhance the paper :

– In the introduction, Early and Late Mesolithic as well as Early Neolithic would deserve a small paragraph to summarize their respective datations, characteristics in terms of type of sites and subsistence strategies. Meso-Neo contexts should be explained for the readers (real mixed levels or undetermined)?

– A brief review of the archaeobotanical data already available in these contexts prior to this study will also be useful.

–Throughout the paper, the terms cereals, oat, etc is questioning as they introduce an ambiguity in the wild versus domestic nature of the plant remains found. I would suggest the authors to be clearer either in the introductory part (ex: the term cereals is used independently of the domestic or wild nature of the caryopses found) or by precising throughout the text if these terms are referring to undet./wild/domestic species. The best solution would probably to keep the latin designation: Poacee or Triticum Sp. This will help the reader clarify exactly the level of precision of the starch grains and more broadly archaeobotanical analysis to discuss the question of domestication. Accordingly, the title and abstract should be revised (the term cereal is not adequate here, speak rather of Poacee e.g.)

– Some larger references to recent starch grains analysis on Mesolithic/Neolithic worlwide and in European contexts would enlarge the scope.

– Concerning the experimental referentials, the authors should precise which species were tested, and if they fit with the ones found in the archaeological record, in order to ensure that the referential used is transposable to prehistoric remains.

– Though all precautions have been taken to avoid any modern contamination of the surfaces of GSTs, no tests have been undertaken concerning contamination by the surrounding soils. These should be discussed to precise the origin of the starch grains: resulting from grinding actions or resulting from taphonomic transfers from the sediments. In the same idea, in table 4 PDm is mentioned; were the starch grains preserved by the concretions? Was the content of the concretions tested?

– The GST section need some additional information. We need more information about the type of tool (pestle, grinder, crusher, etc). This would help the reader to link the type of action and of tools to the starch remains, and to evaluate the different processing technics involved in the respective Mesolithic and Neolithic traditions.

Some precisions should be added to consolidate the presentation of the corpus. Also be careful about the number of tools analyzed, there seems to be contradictory information from one part to another of the text.

We need more information about GST: raw material, localization of the active surfaces on the blank. Surprisingly, macrotraces from the stereoscopic observation are not characterized anywhere. This should be completed in the text and the figures.

Figure 1: in the legend, give dates and precise type of occupation.

Figure 3: in the legend, precise the datation of tools and their typology. Photos are very small to distinguish use-wear traces on their surface. Besides, some plate with Early Mesolithic and Neolithic tools would be interesting to see if there is an evolution in the characteristics of the tools used, in parallel to the evolution of the plants transformed.

Figure 4 (z, Aa) + Bb, Cc: some precisions are necessary: which tool' number (to refer to the Figure 3)? Besides, scale are not correct (only a white suare with no metric indications)

Figure 7: experimental reference or natural one? If experimental explanations about the experimental tests are necessary. This is also not clear in the text itself.

Figure 9: Problem of legend; some number are missing. Again for archaeological GST, precise which one (number, site, type etc); presented this way it is too general and not rigorous.

*Reviewer #3:*

This is an ambitious and interesting study that systematically examines plant microfossils, other debris, and biochemical markers from dental calculus. The central result is that cereal grasses in the tribe Triticeae, as well as a range of other plants, were being consumed prior to the introduction of domesticated species in the study region. This result is sound, and a valuable contribution to our understanding of long-term plant-human interactions in the region, especially outside of what we understand to be the zones of domestication for the main Neolithic crops.

However, I have two main analytical concerns with the manuscript:

1) There is some tendency to associate starch morphotypes with taxonomic groups without clarifying whether diagnostic characters are present to make secure identifications. This is the case with types IV and VI, for Fabaceae and Cornaceae. It is important to clarify whether these groups are being names as examples of the morphotype, or identified as the taxa present, in which case we need more information about the discriminating power of diagnostic characters in these taxa vs others. Table 2 suggests the latter (positive IDs). The interpretation of Aegilops at the genus level is similar, it's lacking the necessary context to assume confidence in the assignment. I think this is particularly important given the role of Aegilops in bread wheat evolution.

2) The chemical analytical methods are not suitably described, only that mass profiles were compared to records at NIST. As a result, it is very difficult to assess whether the interpretations are reasonable in light of analytical specificity. Specificity of chemical biomarkers is an important consideration and has in the past led to archaeological misinterpretations of alkaloids and other compounds due to shared characteristics. Additionally, hordenine is not diagnostic to cereals, and its presence is not robust evidence of cereal caryopsis ingestion. This alkaloid is widespread in plants, and was only first described in Hordeum, not restricted to Hordeum and relatives. The methods refer to analyzing the "pellets left from a metagenomic study" which is as-yet unpublished. We need more detail about the processing leading to the analyzed pellet, i.e. is it an untreated piece of the calculus or the derivative of a DNA isolation procedure, which could have implications for analysis.

These two issues lead to some over-interpretations in my view, which I discuss more in my detailed comments. I think the central contribution is a valuable one, and I would advise the authors to pull back on some of their additional interpretations at this time. I am very appreciative that the authors have undertaken this work in conjunction with meta-genomic analysis for a robust multi-proxy treatment of these important remains, which will allow the most robust possible interpretations to be drawn.

I enjoyed reading this study, as above I feel over-interpretation is the biggest issue. A few detailed comments follow.

47: "verified until now" slightly implies that you are definitely verifying it with this study, changing it "has not yet been verified" would be clearer.

139: Salivary amylase is the most likely interpretation for this damage ("enzymatic digestion"), I would caution against suggesting that this could be interpreted as cultural modification via food preparation.

The numerous starch micrographs of comparative materials (Figure 6 and 7) are not very helpful without indicating diagnostic characters, with citation, and with specific relevance to the text explicated in the figure captions.

242-45: It is not clear if this form is diagnostic to the family Fabaceae, only "typical of" that family. The strong implication is that these can be interpreted in that way, but it is important to make clear whether these structures are found in other taxa, and clarify the confidence level of a Fabaceae interpretation on that basis. Also, Fabaceae contains nearly 20,000 species in all temperate and tropical regions globally, so again I would caution against the implication that these structures in this study are linked with the familiar crop species mentioned.

297: I don't fully understand "daily life crafting activities"

335: Specify "foxtail millet" as the common name for S. italica to avoid confusion with other mentions of millets.

358: This is the first mention of the amyloplast context. This should be introduced and quantified in the Results section, and more information should be given here on the reliability of this context for determining the degree of processing.

387-388: I disagree that the common view is that groups of people without agriculture and domesticated plants are merely foraging opportunistically, I think this is a slightly outdated view. The bulk of domestication and early agriculture/pre-agriculture literature have come to recognize a longterm mutualism of humans and plants even in what we would interpret as wild forms. This doesn't affect the interpretation here in any problematic way and I do think it is an important topic to explore, only that I think the authors are setting up the result against a "common view" that isn't all that common.

410: I would advise against interpreting cultivation on this basis without further exploration of the ecological characters of these species. Several cultigen wild relatives are notoriously weedy, and so in many cases there's a strong possibility that these taxa colonize newly opened landscapes with no human intervention. In either case, cultivation here is an over-interpretation.

411: Chenopodiaceae has been subsumed within Amaranthaceae.

461: The current study does not provide direct evidence for management of wild stands, only their consumption. This interpretation should be contextualized as likely management in light of other literature and ecological expectations, but it should be clear that management is not being tested here.

470-474: This section is a bit unclear, I can't tell what evidence is being interpreted by whom as implausible in the context of the cited study.

544-560: I'm afraid I find this paragraph to be over-interpreted. As mentioned above, the methods of the GC-MS approach are underdeveloped, and the specificity of the results is not clear, therefore without that extra information, interpreting specific compounds without regard to their botanical breadth and chemical lookalikes is not useful. Re omega-3 fatty acids, I think it's a bit of a reach to try and interpret them in an archaeobotanical context when the riverine resources would be such a clear contributor of these compounds.

Figure 8 is not called out in the text, and I'm not sure of its purpose.

[Editors’ note: further revisions were suggested prior to acceptance, as described below.]

Thank you for resubmitting your work entitled "Wild cereal grain consumption among Early Holocene foragers of the Balkans predates the arrival of agriculture" for further consideration by *eLife*. Your article has been evaluated by George Perry (Senior Editor) and two reviewers.

The reviewers appreciated your efforts in addressing their collective previous comments, and they commented that they now agree that the paper presents compelling evidence for the exploitation of wild grasses prior to the introduction of domestic cereals in the Balkans. However, there are some remaining issues as pointed out by one of the reviewers that need to be addressed, as outlined below:

The identification of Type Ia starch grains as a domestic cereal seems to be based on the absence of a bimodal pattern of starch granules distribution in MOST (but not all) wild Aegilops species. Moreover, other wild Triticeae taxa have not been considered as potential origin for these starch grains. For example, Yang and Perry (2013: Figure 2b in JAS) found a bimodal assemblage in Leymus chinensis. This taxon does not naturally occur in the Balkans, but others within the genus do (e.g., Leymus racemosus). Other potential contributors include the genus Bromus, widely distributed throughout Eurasia. For these reasons, I do not think Type Ia should be assigned to a domestic cereal; at most, it can be suggested that it likely does not belong to the genus Aegilops.

---

## [Author Response]

[Editors’ note: the authors resubmitted a revised version of the paper for consideration. What follows is the authors’ response to the first round of review.]

Reviewer #1:This manuscript presents the results of microbotanical (starch grains and phytoliths) and chemical analyses in dental calculus from 61 Mesolithic and Early Neolithic individuals from several sites in the Danube Gorges of the central Balkans. Microbotanical evidence from 44 grinding stones from one of the studied sites is presented as additional evidence supporting the consumption of plants (wild grasses, pulses and fruits) before the arrival of southwest Asian domestic crops.Archaeologists have long acknowledged the use of plants by hunter-gatherer societies worldwide prior to the advent of agriculture. In the Introduction, the authors claim that "forager knowledge and consistent use of wild cereals are still debated and poorly documented outside of the assumed centers of domestication in southwestern Asia" (l. 40-41) and that "the hypothesis of a systematic use of wild cereals (e.g., Aegilops spp.; Hordeum spp.) in this region [the Balkans] during the Mesolithic has not been verified until now" (l. 46-47). These claims seem to justify the novelty of this study. However, the beginning of the Discussion reviews evidence for wild grass use by hunter-gatherers in southeastern Europe, including one of the study sites, in the form of macro and microbotanical remains and pollen grains. The present study, thus, adds to this body of evidence (which should be presented in the Introduction, not the Discussion) in a specific region.

We thank the Reviewer for their comments. As suggested by the reviewer, we have restructured the article so that the part of the discussion mentioned by the reviewer is presented in the Introduction. However, we would like to emphasize that all previous studies that mention wild grains among Mesolithic foragers in the Balkans present circumstantial evidence and the novelty of our findings does not simply “add to this body of evidence” but rather provides the first direct evidence for the use of wild grains among Holocene foragers in the Balkans.

The robustness of this study lies in the high number of samples analysed. However, I do not believe that the analysed datasets are comparable. Dental calculus and grinding stones do provide independent evidence for the consumption of plant resources, but only when analysed from the same chronological periods and the same sites. In this study, dental calculus samples come from six sites dating to the Early Mesolithic, Late Mesolithic, the Mesolithic-Neolithic transition and the Early Neolithic, but samples from grinding stones come from a single site and a single occupational deposit (Late Mesolithic). Moreover, the analyses conducted in each set of samples are not consistent either-microbotanical and chemical analyses in dental calculus vs. microbotanical analyses in grinding stones-further precluding the comparability of the resulting datasets. In my opinion, these samples are not comparable and should not be presented as part of the same study.

We agree with the Reviewer that dental calculus and grinding stones provide independent evidence for the consumption of plant foods when analyzed from the same chronological periods and the same sites. However, we still believe that the two lines of evidence are comparable, complementary, and reinforce the conclusions reached about the use of wild cereal grasses. There is no region of the world where one could expect a perfect preservation and match of two different strands of evidence due to a host of reasons, including taphonomic and other preservation biases and the nature of the record. In the new version of the article, we explained better why we primarily find the category of ground stone artifacts from the Late Mesolithic period onwards, which relates to the observed pattern of only sporadic use of grounds stone in earlier periods in the region of the Danube Gorges and the appearance of these types of tools only in the Late Mesolithic (Antonović et al., 2006; Borić et al., 2014; Srejović and Letica, 1978). Likewise, it would be unrealistic to expect us to analyze all sites and their ground stone assemblages for this study and we believe that a sample of over 100 artefacts suffices in order to reach robust and reliable conclusions. We also believe that the evidence from the site of Vlasac, as one the most representative sites in the Danube Gorges area, characterized by a highly developed Late Mesolithic practice of using ground stones for wild grain cereal processing, reflects wider practices of forager ground stone use across the region.

Regarding the identification of microscopic plant remains in samples from dental calculus and grinding stones, I am puzzled by the identification of some Type I starch grains (Triticeae tribe) as belonging to the genus Aegilops. In a previous study in which several of the authors of the current paper participated (Cristiani et al. 2016), the identification of Triticeae starch grains as Aegilops was specifically disregarded because "Aegilops spp. are absent from assemblages with analyzed macrobotanical remains found at Mesolithic and Early Neolithic sites in the central Balkans" (Cristiani et al. 2016: 4). However, in the present study the authors seem comfortable assigning part of the encountered Triticeae starch grains to this genus without rectifying their previous claim.

We thank the Reviewer for their comment. We have now explicitly rectified our previous claims that “it was possible to exclude wild species of the Triticeae […] that could have been eaten at the time in the region […]. Furthermore, *Aegilops* spp. are absent from assemblages with analyzed macrobotanical remains found at Mesolithic and Early Neolithic sites in the central Balkans.”. We underline that our current conclusion about the consumption of the species of genus Aegilops in the Mesolithic Danube Gorges is not based on the absence of macro remains in the archaeological record but is rather grounded in the analysis of a very large pool of Mesolithic individuals (tot. 61 individuals). In our 2016 study, we included only 9 Mesolithic individuals out of which only 5 yielded Triticeae tribe starch granules. In our current study, we analysed 26 Late Mesolithic individuals out of which 16 yielded Triticeae starch granules; 8 Mesolithic-Neolithic individuals out of which 5 yielded Triticeae tribe starches; and, more importantly, several Early Mesolithic individuals (not included in our previous article) of which 5 yielded Triticeae tribe starches. Hence, we observed a large population of ancient Triticeae starch granules, which made our current conclusions more robust. The analysis of so many individuals also provided a time-depth perspective on the presence of various species. The morphological variability identified in the ancient Triticeae starch population from the Danube Gorges could not be evaluated extensively in 2016 PNAS as only 3 Late Mesolithic individuals yielded starch granules attributed to the Triticeae tribe. Moreover, no Early Mesolithic individuals were analysed at that time. In the dental calculus of the 3 Late Mesolithic individuals discussed in 2016, only one type of ancient Triticeae granules was identified (described in the text as “subtype Ib”), which we attributed to domesticated cereals. Well-known limitations in the inclusion, preservation, and recovery of plant debris in dental calculus could be responsible for the absence of Aegilops starch granules before. However, besides mentioning the absence of Aegilops species in the macrobotanical record of the region, which could also be the result of a host of taphonomic and recovery problems and should not be used to exclude the use of this genus during the Mesolithic (contra Cristiani et al. 2016: 4). In our 2016 paper, we also explicitly claimed that “it was possible to exclude wild species of the Triticeae […] on the basis of their morphology as well as most recent literature on phylogenetic evaluation of Poaceae species”. The morphological criteria used for interpreting ancient Triticeae starch granules haven’t changed in the current work and, indeed, the morphological parameters used in identifying wild (i.e. Aegilops) vs. domesticated species of the Triticeae tribe have only been reinforced, thanks also to newly added statistical analysis. However, we believe that the analysis of a larger subset of individuals, now also including Early Mesolithic individuals, allowed us to observe the morphological variability of the ancient starch population, something that we previously could not fully comprehend given a limited number of individuals analyzed.

Also in regards to the starch assemblage, there seem to be two categories of Type I starch grains, one identified as Aegilops and another as a domestic species within the Triticeae tribe. If the authors believe that this is the case, I recommend separating them into subtypes and treating them a separate taxa for the morphometric analysis.

We thank the Reviewer for this comment. Following this suggestion, we have now separated two subtypes of Type I starch grains. (l.176-223)

Regarding the starch grain morphometric analysis, I am troubled by the fact that the length of Type A and Type B starch grains from modern and archaeological Triticeae spp. seems to have been analysed together, instead of analysing Type A and Type B grains separately. This makes it impossible to compare the results of this study to previously published data (e.g., Henry et al. 2011; Yang and Perry 2013) and potentially biases the morphometric data from archaeological starch grains due to the fact that small starch grains (in this case, Type B) are less likely to survive than large starch grains (Haslam 2004).

We thank the Reviewer for stressing this. We agree to discuss Type A and Type B grains separately so that our results can be compared to already published other data about archaeological Triticeae spp.

Moreover, following the Reviewer’s suggestion, in the “Results” we have made clearer that we have not attempted the identification of starch granules less than 5 μm to avoid misinterpretation of transitory and small storage starch granules (l.171-172) and referred to the work by Haslam 2004**.**

The phytolith assemblage is smaller and it is unclear whether the authors consider it as evidence for food (l. 265-275) or as 'non-dietary remains' (l. 294). In any case, Figure 4w is mentioned as evidence for 'multicellular dendritic structures' (l. 270), but I do not believe that the image shown in Figure 4w can be qualified as a phytolith. Evidence for sedges is also mentioned in l. 298 but not presented elsewhere.

We have now explained the presence of phytoliths in the dental calculus matrix (l. 501-506). We corrected the mistake about the presence of sedges, which has now been fixed in the text.

Overall, I believe that this study presents evidence for the consumption of wild grasses and other plants during the Mesolithic and the transition to the Neolithic in the Danube Gorges of the Balkans, but I do not think that the evidence is properly presented.

We hope that all the corrections and changes we applied to the text according to the Reviewer’s suggestions helped us presenting the archaeological evidence in a much proper way.

I find several of the terms employed by the authors inaccurate or problematic. Perhaps the most paradigmatic is the term 'cereal', used in the title, the Abstract (l. 27) and throughout the manuscript. The term 'cereal' refers to a cultivated plant within the grass family (Poaceae). I do not believe that the authors aim at implying the existence of 'low level food production' (Smith 2001), often referred to as 'cultivation' (e.g., Bowles 2011), in the Mesolithic Danube Gorges, so I recommend the consistent use of the term 'wild cereal' (e.g., l. 33) or the more accurate 'wild grasses'.

We thank the Reviewer for Their comment. We have now used only ‘wild cereal’, ‘wild grasses,’ or ‘grass grains’.

Other terms which I find potentially problematic include 'complex foragers' (e.g., l. 31)-what is a 'complex forager' and how does it compare to a 'simple' (?) forager? –

The term “complex” foragers has been used in archaeological literature for a very long time (e.g. Price, T.D. & J. A. Brown. [ed.] 1985. *Prehistoric hunter-gatherers: The emergence of cultural complexity*. Orlando, FL: Academic Press, Inc) and it primarily refers to ethnographic and archaeological foragers that are sedentary, focused on particularly abundant resources of the area and characterized by complex traits of social and cultural organization. Ethnographic examples comprise foragers of the NW Coast and the best known archaeological examples are the Natufian foragers in the Levant.

and the use of the term 'microfossils' when referring to ancient starch grains, which are not fossilised and should therefore be referred to as 'microremains' or 'microscopic remains'.

We thank the Reviewer for Their comment. We have changed the term “microfossils” into “micro-remains”.

Finally, the term '"forgotten" millets' (l. 478) is used without explaining what is meant by 'forgotten'. If it refers to the fact that the millet species mentioned in the manuscript are no longer consumed in the Balkans, how is it different from Aegilops?

We thank the Reviewer for raising this point and giving us the chance to better explain what we mean by ‘forgotten millets’ in the text. Small millet species (e.g., *Setaria viridis, S. verticillata)* are often referred to as ‘forgotten’ millets to differentiate them from the major species of millets that are generally considered of much economic importance (cf. Madella M., Lancelotti C., and García-Granero J.-J. 2013 Millet microremains—an alternative approach to understand cultivation and use of critical crops in Prehistory. *Archaeol Anthropol Sci* (2016) 8:17–28 DOI 10.1007/s12520-013-0130-y; Weber SA, Fuller DQ (2008) Millets and their role in early agriculture. Pragdhara 18: 69–90).

We agree with the Reviewer that this definition should have been presented in the text, so we have now changed the phrase “Accordingly, in our analysis of dental calculus and GSTs, we have shown that besides cereals, local foragers consumed oat, legumes, “forgotten” millets, acorns, and Cornelian cherries (Figure 4; Tables 1,2)” into “Accordingly, in our analysis of dental calculus and GSTs, we have shown that besides wild grasses, local foragers consumed oat, legumes, minor millet species of the genus Echinochloa and/or Setaria also known as “forgotten millets” (Madella et al., 2016; Weber and Fuller, 2008), acorns, and Cornelian cherries (Figure 4; Tables 1,2).

Reviewer #2:This paper contributes significantly to the debate on the possible intensification of plants exploitation by late foraging populations prior to strict neolithisation. It focuses on a particular kind of botanical remains, starch grains, from two different kind of sources: dental calculus and ground stone tools. The region understudy, the Balkans, is particularly adapted to discuss this question, while as a series of important Mesolithic and Neolithic sites are well excavated in the considered area. The corpus of individuals is important, and allows solid interpretations. The results presented are convincing, and shows the high diversity of plants consumed by foragers. They also bring solid arguments to demonstrate that Late Mesolithic foragers were engaged in the consumption and probably selection of several of the wild plant species that were cultivated in their domestic form during the Neolithic. It keeps open some questions regarding the exact datation of the domestication, at the heart of intense debates among prehistorians.A couple of improvements could be proposed to enhance the paper :– In the introduction, Early and Late Mesolithic as well as Early Neolithic would deserve a small paragraph to summarize their respective datations, characteristics in terms of type of sites and subsistence strategies. Meso-Neo contexts should be explained for the readers (real mixed levels or undetermined)?

We have now added a small paragraph summarizing chronology, characteristics in terms of type of sites, and subsistence strategies in the introduction as suggested by the Reviewer (l.80-107)

– A brief review of the archaeobotanical data already available in these contexts prior to this study will also be useful.

In the introduction (l. 52-69 and l.118-139), we have now provided a review of the extant archaeobotanical data.

–Throughout the paper, the terms cereals, oat, etc is questioning as they introduce an ambiguity in the wild versus domestic nature of the plant remains found. I would suggest the authors to be clearer either in the introductory part (ex: the term cereals is used independently of the domestic or wild nature of the caryopses found) or by precising throughout the text if these terms are referring to undet./wild/domestic species. The best solution would probably to keep the latin designation: Poacee or Triticum Sp. This will help the reader clarify exactly the level of precision of the starch grains and more broadly archaeobotanical analysis to discuss the question of domestication. Accordingly, the title and abstract should be revised (the term cereal is not adequate here, speak rather of Poacee e.g.)

We have now specified when we refer to wild or domestic cereal. Alternatively, we have used “grass grains” referring to wild/domestic cereal or “wild grasses of the Poaceae family”. We have also revised the title and the abstract using “wild cereals”.

– Some larger references to recent starch grains analysis on Mesolithic-Neolithic worlwide and in European contexts would enlarge the scope.

Following the Reviewer’s suggestion, we have now added new references. (l. 434-437)

– Concerning the experimental referentials, the authors should precise which species were tested, and if they fit with the ones found in the archaeological record, in order to ensure that the referential used is transposable to prehistoric remains.

with regards to the different species of the Triticeae tribe used for our archaeological interpretations, we have detailed them in the descriptions of the morphotypes as well as in Figures 6, 7, and 9. For types II, III, and IV, species used as references are mentioned in the description of the morphotype. In figure 8 details the plant species used for experimental activity have also been detailed. We also used already published data. We mentioned this in the description of the morphotypes.

– Though all precautions have been taken to avoid any modern contamination of the surfaces of GSTs, no tests have been undertaken concerning contamination by the surrounding soils. These should be discussed to precise the origin of the starch grains: resulting from grinding actions or resulting from taphonomic transfers from the sediments. In the same idea, in table 4 PDm is mentioned; were the starch grains preserved by the concretions? Was the content of the concretions tested?

The PDM mentioned in Table 4 refers to the alteration observed across the entire surface of GST. The collection of the samples was performed only on the GST surface free of any concretion or with no visible post-depositional alteration so to avoid sampling residues not associated with the actual use of the tool (i.e., contamination, soil concretion).

– The GST section need some additional information. We need more information about the type of tool (pestle, grinder, crusher, etc). This would help the reader to link the type of action and of tools to the starch remains, and to evaluate the different processing technics involved in the respective Mesolithic and Neolithic traditions.

Thank you for highlighting this. Following the Reviewer’s suggestion, we have now added the column “Tool type” in Table 3, indicating the type of tool.

Some precisions should be added to consolidate the presentation of the corpus. Also be careful about the number of tools analyzed, there seems to be contradictory information from one part to another of the text.

We thank the Reviewer for pointing this out. The number of analyzed tools is now consistent throughout the text.

We need more information about GST: raw material, localization of the active surfaces on the blank. Surprisingly, macrotraces from the stereoscopic observation are not characterized anywhere. This should be completed in the text and the figures.

We thank the Reviewer for the comment. Details regarding the GST raw material have now been included in the text (p. 16 lines 341-342). Figure 3 has been edited, indicating the active surface of the tool, its functional area(s), and the activity performed. Figure 8 has been modified by adding microphotographs of the macro traces. Furthermore, a description of the type of surface modification identified at low magnification has been added in the text (p.16 lines 346-349).

Figure 1: in the legend, give dates and precise type of occupation.

As requested, we have now provided information about the sample chronology in the figure 1 legend.

Figure 3: in the legend, precise the datation of tools and their typology. Photos are very small to distinguish use-wear traces on their surface. Besides, some plate with Early Mesolithic and Neolithic tools would be interesting to see if there is an evolution in the characteristics of the tools used, in parallel to the evolution of the plants transformed.

We thank the Reviewer for the comment. Inventory numbers along with the type of tools have now indicated in Figure 3. Also, the functional areas have now been highlighted in the pictures to ease the reader. As for the assemblage chronology, all the GSTs we analyzed refer to the Late Mesolithic occupation of the site of Vlasac, as reported in the caption.

Figure 4 (z, Aa) + Bb, Cc: some precisions are necessary: which tool' number (to refer to the Figure 3)? Besides, scale are not correct (only a white suare with no metric indications)

Following the Reviewer’s comment, in the caption of Figure 4 we have now added the tool inventory number. Also, a metric indication has now been added to the scales.

Figure 7: experimental reference or natural one? If experimental explanations about the experimental tests are necessary. This is also not clear in the text itself.

All of the starch granules presented in Figure 7 are at their natural state (i.e., not processed with a ground stone) from our modern reference collection.

Figure 9: Problem of legend; some number are missing. Again for archaeological GST, precise which one (number, site, type etc); presented this way it is too general and not rigorous.

We thank the Reviewer for spotting this. Figure 9 (now Figure 8) has been edited, and details concerning the inventory number and the type of tool added. We did not add information on the site as all the archaeological GSTs come from Vlasac.

Reviewer #3:This is an ambitious and interesting study that systematically examines plant microfossils, other debris, and biochemical markers from dental calculus. The central result is that cereal grasses in the tribe Triticeae, as well as a range of other plants, were being consumed prior to the introduction of domesticated species in the study region. This result is sound, and a valuable contribution to our understanding of long-term plant-human interactions in the region, especially outside of what we understand to be the zones of domestication for the main Neolithic crops.However, I have two main analytical concerns with the manuscript:1) There is some tendency to associate starch morphotypes with taxonomic groups without clarifying whether diagnostic characters are present to make secure identifications. This is the case with types IV and VI, for Fabaceae and Cornaceae. It is important to clarify whether these groups are being names as examples of the morphotype, or identified as the taxa present, in which case we need more information about the discriminating power of diagnostic characters in these taxa vs others. Table 2 suggests the latter (positive IDs). The interpretation of Aegilops at the genus level is similar, it's lacking the necessary context to assume confidence in the assignment. I think this is particularly important given the role of Aegilops in bread wheat evolution.

We thank the Reviewer for their comments. In the new version of the article, we have changed the description of morphotypes into the following (lines 316-321): “Granules ascribed to this type have been identified in two individuals (1LM, 1M/N) (Table 2). They are characterized by a round 3D morphology and a central hilum, which appears as a wide depression, and no lamellae or facets (Figure 4i, Figure 5n-p). Zarrillo and Kooyman (2006) consider these morphological features diagnostic of some species of drupes and berries. In our sample, starch granules of this morphotype can reach 12 μm of maximum width, which is beyond species of berries and drupes in the Rosaceae family known in the literature (Zarrillo and Kooyman 2006) and in our modern reference (e.g. *Prunus spinosa*). Based on our experimental record, we assign type VI to species of the family Cornaceae (e.g., *Cornus mas* L.) (Figure 7), the remains of which are documented at Vlasac (Filipović et al. 2010).”

We have also re-written the descriptions of Type I, better explaining the diagnostic characters used in the interpretation of starch assigned to Aegilops.

2) The chemical analytical methods are not suitably described, only that mass profiles were compared to records at NIST. As a result, it is very difficult to assess whether the interpretations are reasonable in light of analytical specificity. Specificity of chemical biomarkers is an important consideration and has in the past led to archaeological misinterpretations of alkaloids and other compounds due to shared characteristics. Additionally, hordenine is not diagnostic to cereals, and its presence is not robust evidence of cereal caryopsis ingestion. This alkaloid is widespread in plants, and was only first described in Hordeum, not restricted to Hordeum and relatives. The methods refer to analyzing the "pellets left from a metagenomic study" which is as-yet unpublished. We need more detail about the processing leading to the analyzed pellet, i.e. is it an untreated piece of the calculus or the derivative of a DNA isolation procedure, which could have implications for analysis.

We thank the Reviewer for this comment. Upon reflection and considering the Reviewer’s comments, we decided not to present the results of GCMS in the new version of the article as we performed the analysis only on a select number of individuals.

47: "verified until now" slightly implies that you are definitely verifying it with this study, changing it "has not yet been verified" would be clearer.

Thank you for highlighting this. We have now changed this phrase into “However, the hypothesis of a systematic use of wild grasses of the Poaceae family (e.g., *Aegilops* spp.; *Hordeum* spp.) during the Mesolithic remains to be verified in this region.”

139: Salivary amylase is the most likely interpretation for this damage ("enzymatic digestion"), I would caution against suggesting that this could be interpreted as cultural modification via food preparation.

We agree that enzymatic digestion is likely the main explanation for damage observed in starch granules entrapped in archaeological dental calculus. However, recent experimental use of ground stone tools for processing various grains has resulted in the production of damaged granules. We have mentioned these results as an alternative scenario for explaining the presence of damaged starch granules in Neolithic individuals. Following the Reviewer’s comment and recent experimental results, we have now changed the phrase into the following: “Type A granules appear damaged in few EN individuals, which may be linked to enzymatic digestion (salivary amylase) although plant food processing could also result in starch damage based on experimental results (Ma et al. 2019; Zupancich et al. 2019).”

The numerous starch micrographs of comparative materials (Figure 6 and 7) are not very helpful without indicating diagnostic characters, with citation, and with specific relevance to the text explicated in the figure captions.

We thank the Reviewer for the comment. Each figure and the relevant panel are now cited correctly in the text. A description of the diagnostic features characterizing each of the granule types identified has been added to the text with reference to figures 6 and 7.

242-45: It is not clear if this form is diagnostic to the family Fabaceae, only "typical of" that family. The strong implication is that these can be interpreted in that way, but it is important to make clear whether these structures are found in other taxa, and clarify the confidence level of a Fabaceae interpretation on that basis. Also, Fabaceae contains nearly 20,000 species in all temperate and tropical regions globally, so again I would caution against the implication that these structures in this study are linked with the familiar crop species mentioned.

Many thanks for this comment. To the best of our knowledge, these characteristics are peculiar and diagnostic of starch granules included in the species of the plant family Fabaceae and not found in other taxa. Following the Reviewer’s suggestions, we have slightly changed the phrasing into the following: “All these features are very peculiar and diagnostic of starch granules included in the species of the plant family Fabaceae (Henry et al., 2011), […]. While many edible species of the family Fabaceae grow in the Balkans (e.g., *Vicia sativa, V. hirsuta, V. ervilia, Lathyrus pratensis, L. sylvestris*), an identification to species or genus was not possible due to overlaps in shape and size of starch granules at tribe level, which were observed in our modern reference collection” (l.298-305).

297: I don't fully understand "daily life crafting activities"

We have changed this into “daily life activities”.

335: Specify "foxtail millet" as the common name for S. italica to avoid confusion with other mentions of millets.

Thanks for this suggestion. We have now specified this in the revised version of the article text.

358: This is the first mention of the amyloplast context. This should be introduced and quantified in the Results section, and more information should be given here on the reliability of this context for determining the degree of processing.

We thank the Reviewer for this comment. We have now mentioned the recovery of starch granules in the amyloplast in the results.

387-388: I disagree that the common view is that groups of people without agriculture and domesticated plants are merely foraging opportunistically, I think this is a slightly outdated view. The bulk of domestication and early agriculture/pre-agriculture literature have come to recognize a longterm mutualism of humans and plants even in what we would interpret as wild forms. This doesn't affect the interpretation here in any problematic way and I do think it is an important topic to explore, only that I think the authors are setting up the result against a "common view" that isn't all that common.

We thank the Reviewer for this comment. We agree that our statement (“This perspective is different from commonly held views about only opportunistic use of available plant foods by foragers”) was inaccurate, so we decided to delete it**.**

410: I would advise against interpreting cultivation on this basis without further exploration of the ecological characters of these species. Several cultigen wild relatives are notoriously weedy, and so in many cases there's a strong possibility that these taxa colonize newly opened landscapes with no human intervention. In either case, cultivation here is an over-interpretation.

In the mentioned statement, which is now moved to the introduction (lines 66-69), we were citing an interpretation discussed by Edwards (1989). However, we have deleted the part of the phrase referring to cultivation.

411: Chenopodiaceae has been subsumed within Amaranthaceae.

Many thanks for pointing this out. We have now fixed this in the text.

461: The current study does not provide direct evidence for management of wild stands, only their consumption. This interpretation should be contextualized as likely management in light of other literature and ecological expectations, but it should be clear that management is not being tested here.

We agree with the Reviewer. We have avoided mentioning management in the phrase.

470-474: This section is a bit unclear, I can't tell what evidence is being interpreted by whom as implausible in the context of the cited study.

We have rephrased the section in the following way: “In the same study, the individuals dated to the Early Mesolithic exhibit a high occurrence of starch granules, which we have shown to be compatible with a variety of wild species of the Triticeae tribe (e.g., genus *Aegilops*). The evidence of Aegilops consumption during the Early Mesolithic was disregard as an “implausible pattern” (lines 542-545).

544-560: I'm afraid I find this paragraph to be over-interpreted. As mentioned above, the methods of the GC-MS approach are underdeveloped, and the specificity of the results is not clear, therefore without that extra information, interpreting specific compounds without regard to their botanical breadth and chemical lookalikes is not useful. Re omega-3 fatty acids, I think it's a bit of a reach to try and interpret them in an archaeobotanical context when the riverine resources would be such a clear contributor of these compounds.

We agree with the Reviewer that in the analyzed context, the presence of omega-3 fatty acids could be more easily explained by referring to the consumption of riverine resources rather than plant foods (although hazelnut DNA was recently recovered in the dental calculus of one Late Mesolithic individual from Vlasac – Ottoni et al. 2021). Upon reflection, we decided not to discuss GCMS results in the article as they were carried out only on a small subset of individuals.

Figure 8 is not called out in the text, and I'm not sure of its purpose.

We thank the Reviewer for this comment. We decided to remove this figure.

[Editors’ note: what follows is the authors’ response to the second round of review.]

The reviewers appreciated your efforts in addressing their collective previous comments, and they commented that they now agree that the paper presents compelling evidence for the exploitation of wild grasses prior to the introduction of domestic cereals in the Balkans. However, there are some remaining issues as pointed out by one of the reviewers that need to be addressed, as outlined below:The identification of Type Ia starch grains as a domestic cereal seems to be based on the absence of a bimodal pattern of starch granules distribution in MOST (but not all) wild Aegilops species. Moreover, other wild Triticeae taxa have not been considered as potential origin for these starch grains. For example, Yang and Perry (2013: Figure 2b in JAS) found a bimodal assemblage in Leymus chinensis. This taxon does not naturally occur in the Balkans, but others within the genus do (e.g., Leymus racemosus). Other potential contributors include the genus Bromus, widely distributed throughout Eurasia. For these reasons, I do not think Type Ia should be assigned to a domestic cereal; at most, it can be suggested that it likely does not belong to the genus Aegilops.

We thank the reviewer for this comment.

We would like to stress that the possibility that archaeological *Type Ia* granules could be ascribed to other wild Triticeae taxa has been considered yet excluded based on experimental data, as reported in the discussion.

Species of the genera *Elymus* (e.g., *Elymus caninus*) and *Agropyrum* (e.g., *Agropyrum pungens; A. farctus*) growing in the Balkans have been analyzed and conclusions were published in our article in 2016 (Cristiani et al. 2016). We excluded the possibility that A-type granules from species of these genera could be compared to the archaeological *Type Ia* on morphological grounds. In particular, (a) starch granules extracted from these species are overall smaller than the *Type Ia* starch granules retrieved in the dental calculus of Mesolithic indiivduals in the Danube Gorges, (b) they show a different morphology (generally oval), and (c) starch granules do not show bimodal distribution. We feel these results on the morphology of starch granules from wild Triticeae taxa should be considered reliable as only the plant material from the region under study has been evaluated.

As suggested by the reviewer, we also explored species of the genus *Bromus* as we are aware they could be recognized as a possible contributor to the *Type Ia* population of starch granules from the Mesolithic Danube Gorges. To the best of our knowledge, only species of the Bromideae tribe, and in particular *Bromus catharticus* (Stoddard and Sarker 2000), are known to have a bimodal distribution. However, this species is not native to the Eurasian region. Other species of the genus *Bromus* (e.g., *Bromus tectorium, Bromus secalinus, Bromus alopecuros, Bromus hordeaceus, Bromus arvensis*) have also experimentally been analysed for the presence of starch granules. However, they were excluded based on the morphology, distribution, and dimensions of their granules. In particular, in these species, the granules are smaller than those in our archaeological record (around or below 11 μm). We also considered the genus *Dasypyron* (e.g. *Dasypyron villosum)*, which is not bimodal.

Our conclusions are also supported by previously published data (Stoddard and Sarker 2000). Based on our experimental data, we confidently exclude that species of the genus *Aegilops, and* other wild species of the Triticeae and Bromideae tribes could be the origin of Type Ia granules. Consequently, we support the possibility that *Type Ia* granules should be ascribed to domesticated species of the Triticeae tribe available in the Balkans since ca. 6500 BC.

Finally, to show how wild taxa of the Triticeae tribe mentioned above are different in their morphology, dimensions, and distribution from the archaeological Type Ia, we have supplemented Figure 6 to include also some of the wild Triticeae taxa we experimentally analyzed for the presence of starch granules.